# Impact of meteorology and aerosol sources on PM$_{2.5}$ and oxidative potential variability and levels in China

**Jiemei Liu**[1,2]**, Jesper H. Christensen**[2]**, Zhuyun Ye**[2]**, Shikui Dong**[1]**, Camilla Geels**[2]**, Jørgen Brandt**[2]**, Athanasios Nenes**[3,4]**, Yuan Yuan**[1]**, and Ulas Im**[2]

[1]Key Laboratory of Aerospace Thermophysics, Ministry of Industry and Information Technology, Harbin Institute of Technology, 92 West Dazhi Street, Harbin 150001, China
[2]Department of Environmental Science/Interdisciplinary Centre for Climate Change, Aarhus University, Frederiksborgvej 399, Roskilde, Denmark
[3]Laboratory of Atmospheric Processes and Their Impacts, Ecole Polytechnique Fédérale de Lausanne (EPFL), Lausanne, Switzerland
[4]Center for the Study of Air Quality and Climate Change, Foundation for Research and Technology Hellas (FORTH), Thessaloniki, Greece

**Correspondence:** Yuan Yuan (yuanyuan83@hit.edu.cn) and Ulas Im (ulas@envs.au.dk)

**Abstract.** China has long-term high PM$_{2.5}$ levels, and its oxidative potential (OP) is worth studying as it may unravel the impacts of aerosol pollution on public health better than PM$_{2.5}$ alone. OP refers to the ability of PM$_{2.5}$ to induce oxidative stress (OS). OP and PM$_{2.5}$ are influenced by meteorological factors, anthropogenic emission sources, and atmospheric aging. Although their impact on PM$_{2.5}$ has been studied, OP measurements only recently became available and on a limited scale, as they require considerable technical expertise and resources. For this, the joint relationship between PM$_{2.5}$ and OP for a wide range of meteorological conditions and emission profiles remain elusive. Towards this, we estimated PM$_{2.5}$ and OP over China using the Danish Eulerian Hemispheric Model (DEHM) system with meteorological input from the Weather Research and Forecasting (WRF) model. It was found that higher values of PM$_{2.5}$ and OP were primarily concentrated in urban agglomerations in the central and eastern regions of China, while lower values were found in the western and northeastern regions. Furthermore, the probability density function revealed that about 40 % of areas in China had annual average PM$_{2.5}$ concentrations exceeding the Chinese concentration limit. For OP, 36 % of the regions have OP below 1 nmol min$^{-1}$ m$^{-3}$, 41 % have OP between 1 and 2 nmol min$^{-1}$ m$^{-3}$, and 23 % have OP above 2 nmol min$^{-1}$ m$^{-3}$, which are in line with previous measurement studies. Analysis of the simulations indicates that meteorological conditions contributed 46 % and 65 % to PM$_{2.5}$ concentrations and OP variability, respectively, while anthropogenic emissions contributed 54 % and 35 % to PM$_{2.5}$ concentrations and OP variability, respectively. The emission sensitivity analysis also highlighted the fact that PM$_{2.5}$ and OP levels are mostly determined by secondary aerosol formation and biomass burning.

## 1 Introduction

Fine particulate matter, with an aerodynamic diameter of less than 2.5 µm (PM$_{2.5}$), is the primary atmospheric pollutant in China (Chen et al., 2021; Chen and Cao, 2021; J. Liu et al., 2023) with respect to human health. PM$_{2.5}$ exposure in China for 2017 resulted in an estimated 1.8 (95 % CI: 1.6, 2.0) million premature deaths (M. Liu et al., 2021). Many recent studies have suggested that the oxidative potential (OP) of PM$_{2.5}$ may better explain the negative impact of PM$_{2.5}$ exposure on human health than the well-established metric of mass concentrations (Yu et al., 2019; Gao et al., 2020). OP refers to the ability of PM$_{2.5}$ to induce oxidative stress (OS) (Yang et al., 2021). Liu et al. (2020) summarized OP measurements conducted in nine regions of China around 2014. The results showed that the average OP content in northern Beijing was highest during the winter of 2016 ($\sim 14.0$ nmol min$^{-1}$ m$^{-3}$), while the average OP level in Shanghai during the spring of 2016 was lowest ($\sim 0.15$ nmol min$^{-1}$ m$^{-3}$). However, there is currently no exact threshold division of OP values. Exposure to high levels of OP (from compounds such as quinones and soluble transitional metals) induces an excess production of reactive oxygen species (ROS) in cells and leads to OS effects that ultimately trigger inflammation and disease. Therefore, reducing PM$_{2.5}$ pollution and its associated OP (the volume-normalized dithiothreitol activity) is critical to addressing China's environmental and environmental health issues.

Anthropogenic emissions, as the main source of PM$_{2.5}$ pollution and environmental health risks, have been studied extensively (Chen et al., 2019; Liu et al., 2022a). Zhu et al. (2018) and Pui et al. (2014) summarized the studies on PM sources in China and reported that secondary inorganic aerosols (SIAs), industry, residential combustion, biomass burning, industry, and transportation are the main source categories in China in the historical and future business-as-usual scenarios. Due to the significant influence of various sectors on PM$_{2.5}$ emissions and research (Liu et al., 2018; Liu et al., 2020) indicating a close association between PM$_{2.5}$ and OP, the connection between OP, serving as a toxicity indicator for PM$_{2.5}$, and its sources (Liu et al., 2020) is becoming increasingly crucial and the topic of numerous studies. For instance, Yu et al. (2019) used a dithiothreitol (DTT) assay to measure the PM$_{2.5}$ samples in Beijing throughout the year and identified vehicle emissions as the main contributing source based on the source analysis of OP. However, studies conducted in three coastal cities of the Bohai Sea region (Liu et al., 2018) and in Nanjing (Zhang et al., 2023) using the same DTT assay indicated that coal combustion was the most active source of OP. Together, these studies suggest that obtaining the spatial distribution characteristics of PM$_{2.5}$ and OP as well as their links to emission sources is of paramount importance for implementing region-specific control measures.

Apart from anthropogenic emissions, meteorological conditions (i.e., temperature, humidity, wind speed, precipitation) also play a crucial role in the formation, accumulation, transformation, and dispersion of PM$_{2.5}$ (Liu et al., 2022a, b). Utilizing a multiple linear regression model, Gong et al. (2022) conducted an analysis of the trends of meteorological elements and PM$_{2.5}$ levels across various regions in China from 2013 to 2020. Furthermore, they separated and quantified the impacts of meteorological factors and emissions on these trends. The findings indicate that meteorology alone can account for approximately 20 %–33 % of the variability in PM$_{2.5}$ levels. Xing et al. (2023) conducted a study in the Shenzhen region using DTT, ascorbic acid (AA), and glutathione (GSH) OP assays. They analyzed meteorological conditions and PM$_{2.5}$ chemical composition to understand how the prevalence of monsoons in winter (northern and northeastern winds) and summer (southern and southeastern winds) affected the sources and contributed to the seasonal variation in PM$_{2.5}$ composition and OP (mass-normalized). Similarly, Molina et al. (2023) and Wang et al. (2019) revealed that meteorological conditions indirectly influence OP (volume-normalized and mass-normalized) through their impact on the chemical properties of the components. Ainur et al. (2023), employing a DTT assay, investigated outdoor health risks associated with atmospheric particulate matter in Xi'an and found a positive correlation between winter OP (volume-normalized) and relative humidity. Although several studies have identified linkages between meteorological conditions and PM$_{2.5}$ and OP, quantitative assessment of meteorological conditions for both PM$_{2.5}$ and OP variability is lacking.

As of the present, research on the influence of both meteorological conditions and anthropogenic emissions on OP primarily relies on measurement methods (Yu et al., 2019; Gao et al., 2020; Campbell et al., 2021), such as DTT, AA, and GSH, which are difficult and costly to test and make it hard to provide the spatial distribution of OP comprehensively. Although mechanistic models of OP do exist (Shahpoury et al., 2024), their links to experimental metrics of OP are qualitative. For this, we propose a hybrid approach combining existing observations of OP with a chemistry transport model (CTM). So, OP from assays and their observed links to sources and chemical constituents can then be parameterized and implemented in a CTM for a comprehensive assessment of OP exposure over large areas and time periods.

This study quantifies the contribution of meteorological and anthropogenic emission factors (i.e., coal combustion, biomass burning, secondary aerosol formation that originate from a series of atmospheric reactions, industry, and transportation sources) to OP and PM$_{2.5}$ levels throughout China with the Danish Eulerian Hemispheric Model (DEHM). The study hence provides a method for calculating OP across China and using OP as an indicator to assess the impacts of anthropogenic emission sources on human health in China.

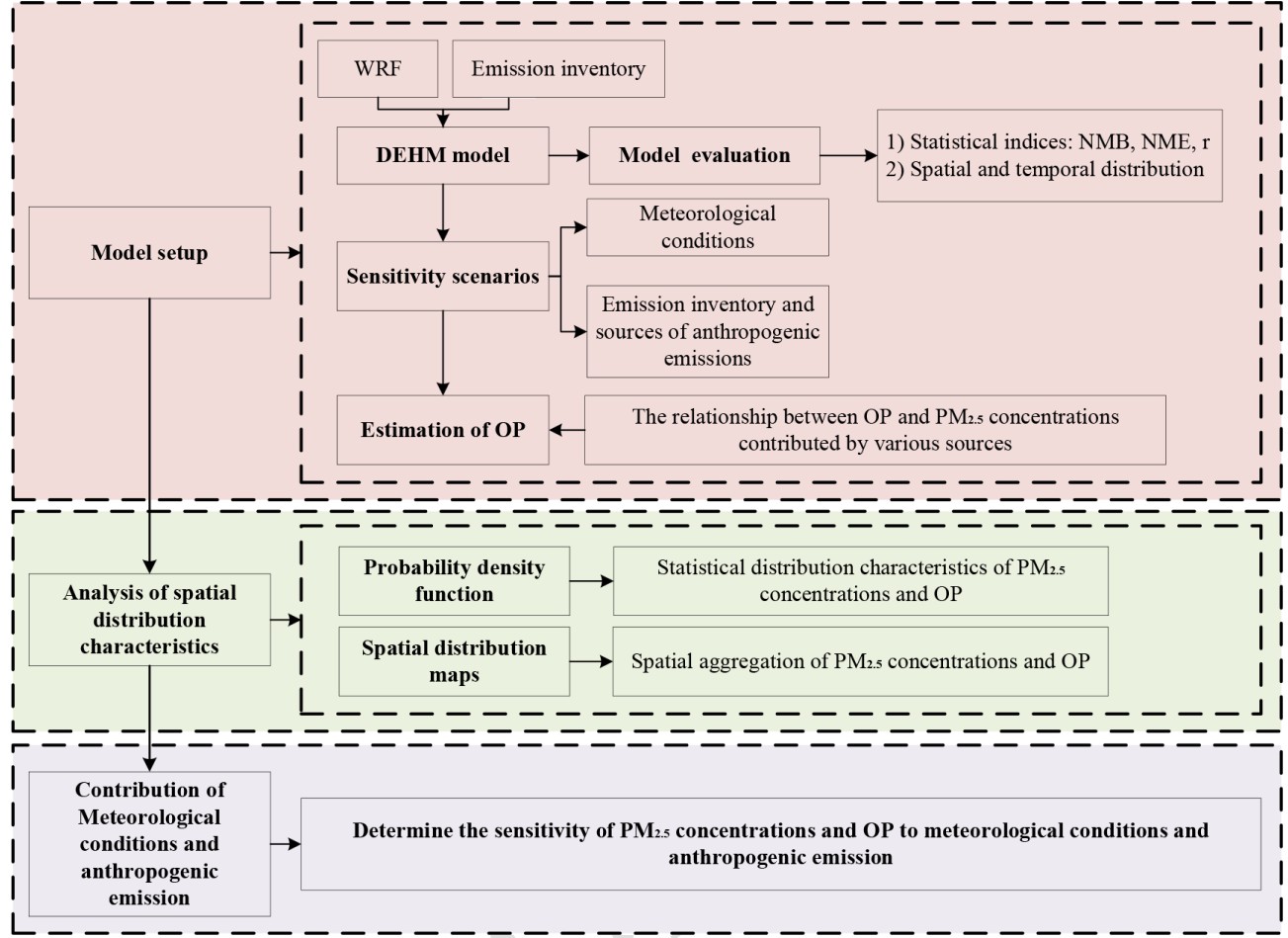

**Figure 1.** Schematic diagram of the study strategy; NME, NMB, $r$, and OP are normalized mean error, normalized mean bias, correlation coefficient, and oxidative potential, respectively.

## 2 Materials and methods

### 2.1 Methodological flow

The research strategy of this study consists of three main parts: model setup, spatial distribution characteristic analysis, and quantification of meteorological conditions and anthropogenic emission contributions (Fig. 1). In the first part, DEHM was employed to obtain hourly pollutant concentrations, followed by model evaluation, where the Weather Research and Forecasting (WRF) model v4.1 (Skamarock et al., 2008) driven by either ERA5 or global climate data from the Community Earth System Model (CESM) was used as meteorological input to DEHM and with exactly same spatial setup for China as in DEHM. Sensitivity experiments were designed for meteorological conditions, emission inventories, and anthropogenic emission sources. From these simulations, the spatial-scale estimation of OP was estimated by incorporating simulated values of primary and secondary PM₂.₅ concentrations from various anthropogenic sources into Eq. (1) (see Sect. 2.2 for details).

In the second part, the spatial distribution characteristics of PM₂.₅ and OP were determined using probability density functions (PDFs) and spatial distribution maps. In the third part, quantitative analysis was conducted based on the simulation results from the sensitivity experiments to determine the extent of the influence of meteorology and emissions on PM₂.₅ and OP, as well as the primary sources of PM₂.₅ and OP.

### 2.2 Estimation of OP

Most current data on OP of PM₂.₅ in China are obtained by means of measurement, and the research objects are basically limited to specific cities, which to some extent hinders conducting research on OP in a large-scale region. Considering that Liu et al. (2018) collected samples across four seasons from multiple representative locations in China, their developed OP prediction model (Eq. 1) can support us in estimating OP (with units of $\mathrm{nmol\,min^{-1}\,m^{-3}}$) in China, thereby exploring the spatial distribution characteristics of OP and the

contributions of different anthropogenic sources to OP. In the present study, we have used this relationship, in combination with the sensitivity simulations (Sect. 2.4), to calculate the OP.

$$OP = 0.088 \times re + 0.076 \times bi + 0.041 \times se$$
$$+ 0.034 \times in + 0.017 \times tr \tag{1}$$

Here, re, bi, in, and tr represent the primary PM$_{2.5}$ concentrations (with units of $\mu g\,m^{-3}$) for coal combustion, biomass burning, industry sources, and transportation sources, respectively. The notation se (secondary aerosol formation) refers to the concentrations of secondary organic and inorganic (SOA and SIA, respectively) components (with units of $\mu g\,m^{-3}$). In this study, coal combustion refers to coal heating from the residential sector. Biomass burning includes open burning of agricultural biomass, domestic biomass burning for cooking and heating, and biomass burning from biomass power plants and coal-fired power plants. The industry source is mainly derived from specific industrial processes in iron and steel, metallurgical production plants for non-ferrous metals (e.g., titanium and molybdenum), and so on. The transportation source primarily comes from tailpipe emissions. It is worth mentioning that secondary aerosol formation originates from a series of atmospheric reactions. Some identified sources (i.e., coal combustion, biomass burning, industrial processes, and transportation) may generate secondary inorganic and organic aerosols through the emission of their precursor components. The coefficient (with units of $nmol\,min^{-1}\,m^{-3}\,source^{-1}$) reflects the intrinsic OP of each source.

## 2.3  Model setup

The DEHM can capture many features of PM well and its precursors' changes in large-scale space (Christensen, 1997; Brandt et al., 2012; Im et al., 2019). To date, the DEHM has been widely used in air pollution and health risk assessment research in Europe and Asia (Brandt et al., 2013a b; Zare et al., 2014; Geels et al., 2015, 2021; Im et al., 2018, 2019, 2023; Lehtomäki et al., 2020; Cramer et al., 2020; S. Liu et al., 2021; Thomas et al., 2022), but this will be the first time that DEHM is applied to estimate OP. Thus, the DEHM system was used to simulate the pollutant concentrations in 2014 by using a two-way nested domain in this study (Kumar et al., 2016). A parent domain with a resolution of $150\,km \times 150\,km$ was employed on a polar stereographic projection, true at $60°\,N$ to cover the entire Northern Hemisphere. The nested domain covered the whole of China, consisting of $150 \times 150$ grid cells with a resolution of $50\,km \times 50\,km$, which was used for the analysis. The mother domain provided initial and boundary conditions for the nested domain. Vertically, there were 29 unevenly distributed layers, with the highest level reaching 100 hPa, and the lowest layer was approximately 20 m in height. The meteorological fields were simulated using the WRF model (Skamarock

et al., 2008) with the same domain and resolution driven by global meteorological data obtained from the ERA5 dataset and the CESM global model, respectively. The simulations utilized the revised MM5 surface layer scheme, the Yonsei University (YSU) boundary layer parameterization scheme (Hong et al., 2006), the multi-scale Kain–Fritsch cumulus parameterization scheme (Zheng et al., 2016), the CAM longwave radiation scheme, and the CAM shortwave radiation scheme (Skamarock et al., 2021). The gas-phase chemistry module included 66 species, nine primary particles (including natural particles such as sea salt), and 138 chemical reactions and was based on the scheme by Strand and Hov (1994) (Brandt et al., 2012). The gas-phase species considered in this study included $SO_2$, $NO_2$, $CH_4$, and $C_2H_6$. PM$_{2.5}$ was formed by BC, OC, sea salt, ammonium ($NH_4^+$), nitrate ($NO_3^-$), sulfate ($SO_4^{2-}$), and secondary organic aerosols (SOAs), among others (Frohn et al., 2022). Biogenic volatile organic compounds (BVOCs), such as isoprene, contributed to the formation of SOA (Zare et al., 2012). Further details on the configuration of the chemical scheme and the list of chemical reactions are in the literature (Zare et al., 2012; Brandt et al., 2012; Collin, 2020; Frohn et al., 2022). The SOAs were calculated using the volatility basis set (see details in Im et al., 2019). In addition to the anthropogenic emissions, DEHM also includes emissions from biogenic emissions, such as vegetation, sea salt, lightning, and soil. The current version of the DEHM does not include windblown resuspended dust emissions, road dust, or aerosol–radiation or radiation–cloud interactions. The time resolution of the DEHM output is 1 h.

In the current study, the DEHM used anthropogenic emissions from the Emissions Database for Global Atmospheric Research – Hemispheric Transport of Air Pollution (EDGAR-HTAP) and Eclipse V6. The biomass burning emissions are obtained from the Global Fire Assimilation System (GFAS) from ECMWF (Kaiser et al., 2012), which has a horizontal resolution of $0.1° \times 0.1°$ on a daily time basis. Natural emissions for DEHM are based on the Global Emissions InitiAtive (GEIA, Zare et al., 2012; Frost et al., 2013) with monthly inventories for emissions of nitrogen oxides from soil and lightning and annual inventories for emissions of ammonia from natural sources. The production of sea salt (Soares et al., 2016) and biogenic volatile organic compounds (Zare et al., 2014) is calculated online in the model as a function of meteorological parameters like wind speed and temperature (Frohn et al., 2022).

## 2.4  Sensitivity scenarios

### 2.4.1  Relative contributions from meteorological conditions and emissions

Table 1 summarizes the scenarios for assessing the relative contributions of meteorological conditions and emissions to PM$_{2.5}$ and OP variability in 2014. ERA5 (Hersbach et al.,

**Table 1.** Emission inventory and meteorological datasets in three simulation scenarios.

| Scenarios | Emission inventory | Meteorological datasets |
|-----------|-------------------|------------------------|
| C$_1$ | EDGAR-HTAP | ERA5 |
| C$_2$ | Eclipse V6 | ERA5 |
| C$_3$ | Eclipse V6 | CESM |

2020; ERA, 2023) is a global reanalysis dataset that is based on the assimilation of historical observations and model data. Studies (Thomas et al., 2021; Xu et al., 2022) have demonstrated that ERA5 performs well relative to MERRA, NCEP, and ERA-Interim, with higher temporal and spatial resolutions. Therefore, scenarios C$_1$ and C$_2$ used ERA5 as input to WRF. Considering the robust representation of aerosol effective radiative forcing and good predictive capabilities for key surface variables in CESM (2023) (García-Martínez et al., 2020; Richter et al., 2022), scenario C$_3$ utilized meteorological data based on CESM version 2.1.1 (Danabasoglu et al., 2020) as input for WRF. Scenarios C$_2$ and C$_3$ employed the Eclipse V6 emissions inventory, while scenario C$_1$ used the EDGAR-HTAP inventory.

The ECLIPSE project by the International Institute for Applied Systems Analysis (IIASA) aims to generate a global gridded anthropogenic emission inventory for various emission scenarios. The Greenhouse Gas – Air Pollution Interactions and Synergies (GAINS) model has been employed to estimate emissions of air pollutants and GHGs (such as SO$_2$, NO$_x$, NH$_3$, NMVOC, BC, OC, OM, PM$_{2.5}$, PM$_{10}$, CO, and CH$_4$) using source characteristics and emission factors at a resolution of $0.5° \times 0.5°$ latitude–longitude (Upadhyay et al., 2020; Eclipse, 2020). The following sectors are available: energy, industry, solvent use, transport, domestic combustion, agriculture, open burning of agricultural waste, and waste treatment. A number of scenarios are provided for which the key economic assumptions and energy use originate from the IEA World Energy Outlook (IEA, 2011), the POLES model, or Energy Technology Perspectives (IEA, 2012) for the period 2010–2050, while statistical data for the period 1990–2010 came from IEA. For agriculture the FAO databases and long-term global projections were used (Alexandratos and Bruinsma, 2012). It is noteworthy that this inventory considers China's 13th 5-year plan. The EDGAR-HTAP (Joint et al., 2011; Crippa et al., 2023) emission inventory endeavors to employ official or scientific inventories within a national or regional scale, with a spatial resolution of $0.1° \times 0.1°$. EDGAR-HTAP comprehensively accounts for all major emission sectors, including residential, transportation, industrial, energy, and agricultural sectors. EDGAR offers independent emission estimates for various pollutants, including CO, CH$_4$, SO$_2$, NO$_x$, NMVOCs, NH$_3$, PM$_{2.5}$, PM$_{10}$, BC, and OC. These estimates follow a standardized methodology provided by the Intergovernmental

Panel on Climate Change (IPCC). The data from EDGAR allow for comparisons with emission reports published by European Member States or Parties under the United Nations Framework Convention on Climate Change (UNFCCC) (Kumar et al., 2023).

Equations (2)–(5) were used to quantitatively evaluate the contributions of meteorological conditions and emission inventories.

$$\mathrm{Con}(\mathrm{Met}) = \frac{C_2 - C_3}{C_3} \qquad (2)$$

$$\mathrm{Con}(\mathrm{Emi}) = \frac{C_1 - C_2}{C_2} \qquad (3)$$

$$\mathrm{NCon}(\mathrm{Met}) = \frac{\mathrm{abs}(\mathrm{Con}(\mathrm{Met}))}{\mathrm{abs}(\mathrm{Con}(\mathrm{Met})) + \mathrm{abs}(\mathrm{Con}(\mathrm{Emi}))} \qquad (4)$$

$$\mathrm{NCon}(\mathrm{Emi}) = \frac{\mathrm{abs}(\mathrm{Con}(\mathrm{Emi}))}{\mathrm{abs}(\mathrm{Con}(\mathrm{Met})) + \mathrm{abs}(\mathrm{Con}(\mathrm{Emi}))} \qquad (5)$$

$C_1$, $C_2$, and $C_3$ represent the PM$_{2.5}$ concentrations and OP from scenarios C$_1$, C$_2$, and C$_3$, respectively. Con(Met) represents the impact of changing meteorological datasets on changes in PM$_{2.5}$ and OP. Con(Emi) represents the impact of changing emission inventory on changes in PM$_{2.5}$ and OP. NCon(Met) and NCon(Emi) represent the normalized contributions of meteorology and emissions. In the equations, the abs function represents the absolute value of the quantity in parentheses.

### 2.4.2 Relative contributions from individual emissions

As mentioned above, OP's main source contributions include five categories, i.e., coal combustion, biomass burning, secondary aerosol formation, industrial sources, and transportation sources (Eq. 1). We conducted perturbation experiments targeting these five sources to quantitatively assess their contributions to PM$_{2.5}$ concentrations and OP (Fig. 2). These experiments were carried out within the three scenarios described in Sect. 2.4.1, and we performed a total of 15 runs. Under the non-perturbation condition (referred to as the NPC case), all aforementioned emission sources were considered. Under the perturbation condition (referred to as the PC case), reduction designs were implemented for emissions from coal combustion, biomass burning, industrial sources, and transportation sources. The emissions of both primary aerosols and trace gases from each individual source are reduced by 30 %. The choice of 30 % was motivated by the consideration that the perturbation would be large enough to produce a sizable impact (i.e., more than numerical noise) even at long distances while being small enough to be in the near-linear atmospheric chemistry regime (Galmarini et al., 2017; Im et al., 2019). Notably, the transportation sector in the DEHM only considers tailpipe emissions, excluding non-exhaust emissions from vehicles like road dust, brake dust, and tire wear. The contribution of tailpipe emissions in the transportation sector to OP is estimated by incorporating simulated values of primary PM$_{2.5}$ concentrations from tailpipe

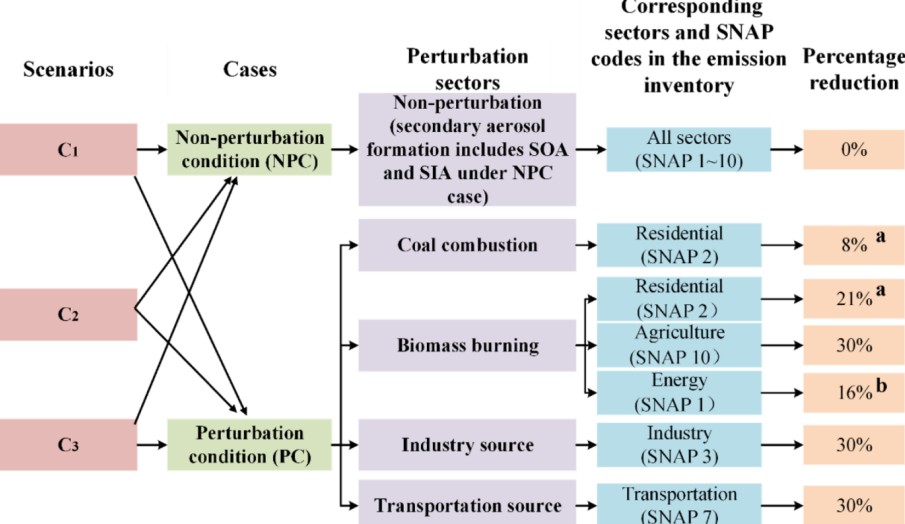

**Figure 2.** Emission reduction design for perturbed emissions **(a)** obtained from Yun et al. (2020) and **(b)** obtained from additional literature (Zheng et al., 2018; Tong et al., 2018; Yun et al., 2020; MEE, 2020; Wang et al., 2020; Tang et al., 2020; Lin et al., 2021; Chen et al., 2022).

emissions into Eq. (1). In this study, the transportation sector refers to road transport, excluding other transportation sources like ships and airplanes. Additionally, to estimate the PM$_{2.5}$ concentrations and OP from coal and biomass burning, it is necessary to obtain the percentage contributions of PM$_{2.5}$ emissions from coal combustion for residential heating; domestic biomass burning for cooking and heating to PM$_{2.5}$ emissions of the residential sector, respectively; and the percentage contributions of PM$_{2.5}$ emissions from biomass burning in power plants to the total PM$_{2.5}$ emissions from the power sector. The percentage contributions of each anthropogenic source can be estimated using Eqs. (6)–(8).

$$PC_{re\_j} = \frac{E_{re\_j}}{E_{re}} \tag{6}$$

$$E_{pp\_bi} = EF \times FQ \tag{7}$$

$$PC_{pp\_bi\_cf} = \frac{E_{pp\_bi} + E_{pp\_cf}}{E_{pp}} \tag{8}$$

$PC_{re\_j}$ denotes the percentage contribution of PM$_{2.5}$ emissions from the residential subsector $j$ (including coal cooking, coal heating, biomass cooking, biomass heating, clean energy, and nonresidential) to the total PM$_{2.5}$ emissions from the residential sector. $E_{re\_j}$ represents the PM$_{2.5}$ emissions from the residential subsector $j$, while $E_{re}$ represents the total PM$_{2.5}$ emissions from the residential sector. The values of $E_{re\_j}$ and $E_{re}$ are obtained from the literature (Yun et al., 2020). After calculation, the values for PC$_{re_{coal\ cooking}}$, PC$_{re_{coal\ heating}}$, PC$_{re_{biomass\ cooking}}$, and PC$_{re_{biomass\ heating}}$ are determined to be 21 %, 27 %, 33 %, and 19 %, respectively. $E_{pp\_bi}$ refers to the PM$_{2.5}$ emissions from biomass power plants, EF refers to the PM$_{2.5}$ emission factor of biomass power plants, and FQ refers to the fuel quantity. PC$_{pp\_bi\_cf}$ refers

to the percentage contribution of PM$_{2.5}$ emissions from biomass power plants and coal-fired power plants to the PM$_{2.5}$ emissions of the power plants. To accelerate carbon reduction in coal-fired power generation, the Chinese government has issued a series of policies supporting and encouraging the coupling of coal and biomass for power generation (Mao, 2017). This undoubtedly adds complexity to distinguishing between PM$_{2.5}$ emissions from coal combustion and biomass fuel. Furthermore, in the energy strategy where biomass serves as a clean alternative to fossil fuels, the scale of biomass utilization and the biomass power generation industry in China continue to expand (Lin et al., 2021). Considering the aforementioned reasons, we include PM$_{2.5}$ emissions from coal-fired power plants in our analysis. $E_{pp\_cf}$ refers to the PM$_{2.5}$ emissions from coal-fired power plants, and $E_{pp}$ refers to the PM$_{2.5}$ emissions from power plants. EF, FQ, $E_{pp\_cf}$, and $E_{pp}$ are obtained from the literature (Zheng et al., 2018; Tong et al., 2018; Yun et al., 2020; MEE, 2020; Wang et al., 2020; Tang et al., 2020; Lin et al., 2021; Chen et al., 2022). After calculation, PC$_{pp\_bi\_cf}$ is determined to be 54 %. Previous studies (Hodan and Barnard, 2004; Chen et al., 2018; Zhang et al., 2022) showed that in China, the proportion of secondary and primary PM$_{2.5}$ mass to the total PM$_{2.5}$ mass is close, so we assume that they account for 50 %, respectively. Figure 2 shows the emission reduction design for perturbed emissions.

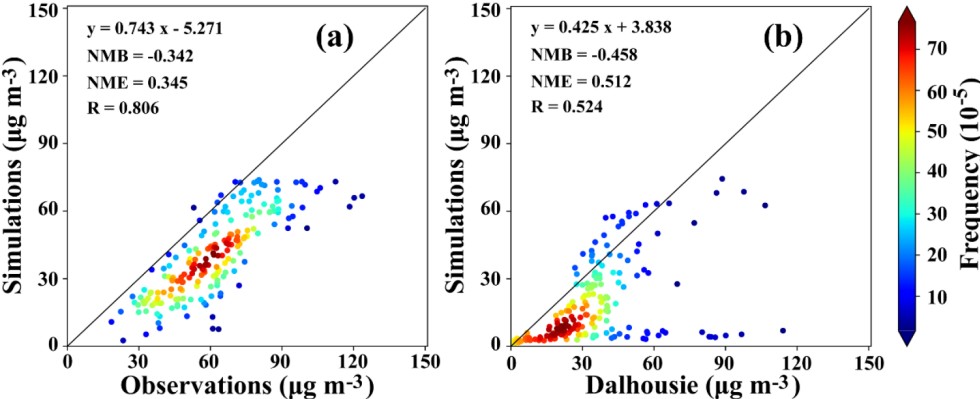

**Figure 3.** Density scatterplots of model performance and validation for China in scenario C$_1$ based on **(a)** annual mean PM$_{2.5}$ observations from MEE and **(b)** annual mean PM$_{2.5}$ derived from the Dalhousie dataset in 2014.

Furthermore, the PM$_{2.5}$ concentrations for each sector are calculated using Eqs. (9)–(11).

$$C_{P,i} = \frac{C_{NPC,i} - C_{PC,i}}{30\%} \tag{9}$$

$$C_{P,\text{primary PM}_{2.5}} = C_{P,\text{total PM}_{2.5}} - C_{P,SIA} - C_{P,SOA} \tag{10}$$

$$C_{\text{secondary}} = C_{NPC,SOA} + C_{NPC,SIA} \tag{11}$$

Here, $i$ refers to the type of pollutants, i.e., total PM$_{2.5}$, SOA, SIA, and primary PM$_{2.5}$. $C_{NPC,i}$ represents concentrations of the pollutant $i$ in the NPC case. $C_{PC,i}$ represents concentrations of the pollutant $i$ in the PC case. $C_{P,i}$ represents concentrations of the pollutant $i$ by the specific emission sector P, which is perturbed (perturbation sectors P include coal combustion, biomass burning, industry, and traffic sources). $C_{P,\text{primary PM}_{2.5}}$ represents concentrations of primary PM$_{2.5}$ by the perturbation sector P. $C_{\text{secondary}}$ represents PM$_{2.5}$ concentrations from secondary aerosol formation.

## 2.5 Probability density function

Taking into account the substantial spatial heterogeneity of PM$_{2.5}$ concentrations and OP, we employ PDF to characterize the statistical distribution characteristics of PM$_{2.5}$ concentrations and OP across China. This offers a more generalized and robust probability for criteria limits. In this study, all three functional types (lognormal, exponential, and gamma) were tested for annual average PM$_{2.5}$ concentrations and OP at the monitoring stations. To determine the representative distributions for the datasets, we further performed goodness-of-fit tests such as the sum of squared error (SSE) and the Kolmogorov–Smirnov (K-S) test (de Melo et al., 2000) using the fitter package in Python.

## 3 Results and discussion

### 3.1 Model evaluation

The hourly observation data were obtained from the Ministry of Ecology and Environment of China (MEE, 2014). The MEE website first released PM$_{2.5}$ measurement data in January 2013. In accordance with Chinese environmental protection standards, the hourly PM$_{2.5}$ concentrations are measured using the micro-oscillation balance method and beta absorption method, with an uncertainty of less than 5 µg m$^{-3}$ (Zeng et al., 2021). The PM$_{2.5}$ monitoring stations are primarily distributed in urban areas, particularly in major metropolitan areas of China (Zeng et al., 2021). In 2014, the observation stations were mainly concentrated in eastern China, while stations in western China were limited. Therefore, in the present study, we also evaluated with the gridded annual mean global reanalysis Dalhousie surface PM$_{2.5}$ dataset (van Donkelaar et al., 2021), which combines satellite retrievals of aerosol optical depth, chemical transport modeling, and ground-based measurements. The Dalhousie dataset compensated for the nonuniform distribution spatially of observation stations to comprehensively evaluate the performance of the DEHM. The density scatterplot of model performance and evaluation for China in scenario C$_1$ based on annual mean PM$_{2.5}$ observations from MEE and PM$_{2.5}$ derived from the Dalhousie dataset are shown in Fig. 3. Overall, the model performance in terms of correlation coefficient ($R$) and normalized mean error (NME) calculated based on annual mean observations met the performance criteria suggested by Emery et al. (2017) (NME < 0.5, $R > 0.4$), and the normalized mean bias (NMB) was also close to the performance criteria suggested by Emery et al. (2017) (NMB < ± 0.3). Compared to the observations, the model performance in terms of $R$, NME, and NMB calculated based on the Dalhousie dataset was slightly poorer than but still close to the performance criteria suggested by Emery et al. (2017). Additionally, this study evaluated the

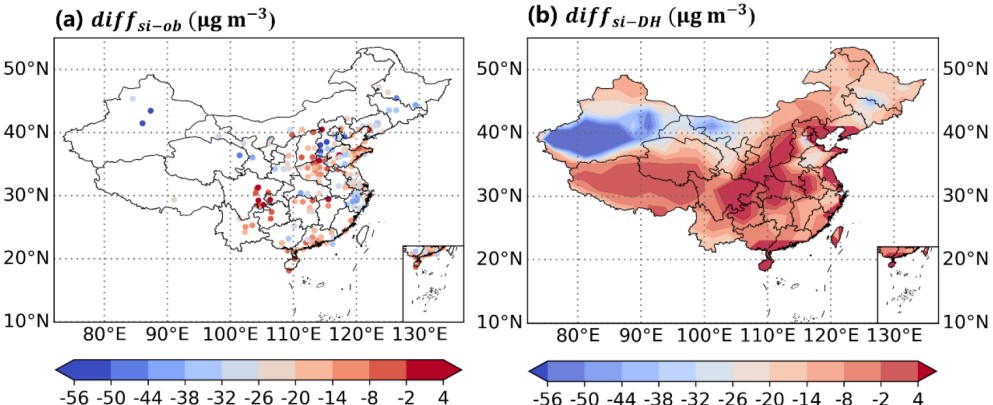

**Figure 4.** Spatial distribution of the annual mean simulated minus annual mean observed values **(a)**, as well as the spatial distribution of the annual mean simulated values minus the Dalhousie dataset **(b)** for China in 2014 under scenario C$_1$. CE1 Publisher's remark: please note that the above figure contains disputed territories.

model performance in scenarios C$_2$ and C$_3$ (Fig. S2 in the Supplement). It was found that under scenarios C$_2$ and C$_3$, the model performance in terms of $R$ and NME, calculated based on both annual mean observations and the Dalhousie dataset, met the performance criteria suggested by Emery et al. (2017). The NMB under scenarios C$_2$ and C$_3$ calculated based on both annual mean observations and the Dalhousie dataset were also close to the performance criteria suggested by Emery et al. (2017). Therefore, the simulated annual mean PM$_{2.5}$ concentrations in scenarios C$_1$, C$_2$, and C$_3$ are considered reliable.

To verify the spatial accuracy, a comparison of the spatial distribution of simulated and observed PM$_{2.5}$, from both MEE and Dalhousie, was conducted. Figure 4 shows the spatial distribution of the annual mean simulated minus annual mean observed values (denoted as diff$_{si-ob}$) (Fig. 4a), as well as the spatial distribution of the annual mean simulated values minus the Dalhousie dataset (denoted as diff$_{si-DH}$) (Fig. 4b). Both Fig. 4a and b indicate that the majority of regions (central and eastern China) exhibited differences ranging from $-18$ to $0\,\mu\mathrm{g\,m^{-3}}$, which is an underestimation of 37 % compared to the average annual observations. The simulated PM$_{2.5}$ concentrations in eastern, central, northeastern, and western China were 37 %, 21 %, $-49$ %, and 41 % lower than the observations, respectively; the simulated values were 28 %, 3 %, 54 %, and 48 % higher than the Dalhousie dataset, respectively. The disparities in model performance across regions may be attributed to uncertainties in the simulation of meteorological fields, coupled with insufficient consideration of species in the reaction processes within the model. Considering the existing literature (Huang et al., 2021; Jia and Zhang, 2021), it is known that bias within approximately 50 % is acceptable. For example, the PM$_{2.5}$ concentrations in eastern China in 2014 simulated by Jia and Zhang (2021) were overestimated by 48 %. Shi et al. (2021) also reported PM$_{2.5}$ concentrations being overesti-

mated or underestimated by 40 % compared to observed values. Hence, the simulated bias in this study falls within an acceptable range, meeting the research requirements.

Similarly, the model performance over timescales was also investigated. Density scatterplots and distribution characteristics of monthly average observations and simulations for all monitoring sites in 2014 are depicted in Fig. S1 in the Supplement and Fig. 5, respectively. From Fig. 5, it can be observed that the simulated values closely align with the observed values from April to September. However, in other months, there was a slightly poorer alignment between simulated and observed values. Nonetheless, considering the overall performance throughout the year, as analyzed in conjunction with Fig. S1, it can be deduced that both the correlation $R$ and NME met the performance criteria suggested by Emery et al. (2017) for all months except December. Furthermore, the results in Fig. 4 indicate that the bias across various regions in DEHM is acceptable. Consequently, on an aggregate level for China, the model demonstrates acceptable performance in simulating monthly average PM$_{2.5}$ concentrations.

## 3.2 Spatial distribution characteristics of PM$_{2.5}$ and OP

To learn about the spatial distributions of PM$_{2.5}$ concentrations and OP, we plotted maps of surface PM$_{2.5}$ and OP for scenario C$_1$ (Fig. 6a and b) and quantified the average annual PM$_{2.5}$ concentrations and OP across different regions of China (Fig. 6c). Figure 6d depicts the geographical location of the study area. High PM$_{2.5}$ concentrations and high OP are mainly located in central and eastern urban clusters. Low PM$_{2.5}$ concentrations and low OP are mainly distributed in northeastern and western China. The results in Fig. 6c indicate that the annual average PM$_{2.5}$ concentrations and OP in eastern, central, northeastern, and western China are $33\,\mu\mathrm{g\,m^{-3}}$ and $1.4\,\mathrm{nmol\,min^{-1}\,m^{-3}}$, $46\,\mu\mathrm{g\,m^{-3}}$ and

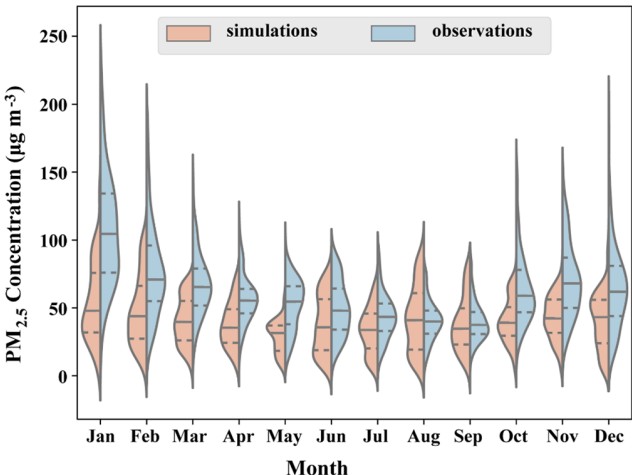

**Figure 5.** Violin plots of the monthly average from MEE observations and simulations averaged over various observation stations for China in 2014 under scenario C$_1$. Red and blue represent the statistical distribution of simulations and observations, respectively. The width of the violin represents the sample size, and the solid black line inside the violin indicates the median. The upper and lower dashed black lines within the violin indicate the upper quartile (the 75th percentile) and lower quartile (the 25th percentile), respectively.

2.0 nmol min$^{-1}$ m$^{-3}$, 19 µg m$^{-3}$ and 0.8 nmol min$^{-1}$ m$^{-3}$, and 12 µg m$^{-3}$ and 0.5 nmol min$^{-1}$ m$^{-3}$, respectively.

Due to differences in city types, pollutant emission intensities, and pollutant chemical components in different regions, there is significant spatial heterogeneity in PM$_{2.5}$ concentrations and therefore in OP (see Sect. 3.4 and Fig. 11 for details). Due to high population density, socioeconomic activities, and winter heating needs, large quantities of anthropogenic emissions, especially from industry, transportation, coal burning, and biomass burning, exacerbate PM$_{2.5}$ and redox-active component pollution.

To quantitatively analyze the regional distribution characteristics of PM$_{2.5}$ concentrations and OP in China, we determined the distribution function that is suitable for a specific dataset (Table 2), investigated the frequency histogram (FH) of PM$_{2.5}$ concentrations and OP, fitted the PDF, and then obtained the cumulative distribution function (CDF) by integrating PDF, as shown in Fig. 7. It was found that the gamma distribution performed the best in fitting PM$_{2.5}$ concentrations and OP from Table 2. Considering the test results, the gamma distribution was used to explore the spatial distribution characteristics of PM$_{2.5}$ concentrations and OP. Figure 7a depicts the probability distribution of PM$_{2.5}$ concentrations, while Fig. 7b depicts the probability distribution of OP. The wide distribution interval indicates that both PM$_{2.5}$ concentrations and OP have a similar and large spatial heterogeneity. According to the FH, the highest frequency density of PM$_{2.5}$ concentrations ranges from 10.5 to 12.9 µg m$^{-3}$; the maximum frequency density of OP ranges

from 0.26 to 0.34 nmol min$^{-1}$ m$^{-3}$. This reflects the overall pollution levels of PM$_{2.5}$ and OP in the Chinese region. Taking into account the annual average PM$_{2.5}$ concentration limits set out in China's ambient air quality standard (AAQS, 2012), we focused on primary (15 µg m$^{-3}$) and secondary concentration (35 µg m$^{-3}$) limits. The PDF and CDF results showed that 85 % of the total area was above the primary concentration limit and 40 % was above the secondary concentration limit. In addition, 36 % of regions in China have an OP below 1.00 nmol min$^{-1}$ m$^{-3}$, 41 % have an OP between 1.00 and 2.00 nmol min$^{-1}$ m$^{-3}$, and 23 % have an OP above 2.00 nmol min$^{-1}$ m$^{-3}$.

### 3.3 Contributions of meteorological conditions and emission inventories to the variations in PM$_{2.5}$ and OP

To determine the sensitivity of PM$_{2.5}$ pollution and oxidation potential (OP) to meteorological conditions (emission inventories), this study compared scenarios C$_2$ and C$_3$ (C$_1$) and investigated the impacts and contributions from ERA5 and CESM (HTAP and Eclipse V6 emission inventories) regarding PM$_{2.5}$ and OP. Figure 8 illustrates the spatial distribution maps of PM$_{2.5}$ concentrations and OP under scenarios C$_1$, C$_2$, and C$_3$. Figure 9a presents the annual average PM$_{2.5}$ concentrations and OP under different scenarios, and Fig. 9b shows the relative contributions of meteorological conditions and emission inventories. From Figs. 8 and 9, it can be observed that, compared to scenario C$_2$, PM$_{2.5}$ concentrations and OP are lower in the western region and slightly higher in some eastern areas under scenario C$_1$, primarily due to changes in emission inventories attributed to the inclusion or exclusion of specific local sources during the compilation process. Compared to scenario C$_3$, PM$_{2.5}$ concentrations and OP are lower in the western region and higher in some eastern areas under scenario C$_2$, primarily attributed to meteorological contributions. For the entire China region, the transition in emission inventories from Eclipse V6 to HTAP resulted in an overall decrease in PM$_{2.5}$ concentrations of 1.55 µg m$^{-3}$, which is approximately 7.61 %, and a decrease in OP of 0.0339 nmol min$^{-1}$ m$^{-3}$, which is approximately 4.05 %. The shift in meteorological data from CESM to ERA5 led to an increase in PM$_{2.5}$ concentrations of 1.22 µg m$^{-3}$, which is approximately 6.4 %, and an increase in OP of 0.0585 nmol min$^{-1}$ m$^{-3}$, which is approximately 7.5 %. According to the normalization process using Eqs. (4) and (5), meteorological conditions contributed approximately 45.6 % to the variations in PM$_{2.5}$ and approximately 65.0 % to the variations in OP. Meanwhile, emission inventories contributed approximately 54.4 % to the variations in PM$_{2.5}$ and approximately 35.0 % to the variations in OP. Our findings highlight the significance of the quality of model input data, including emission inventories and meteorological data, for model performance.

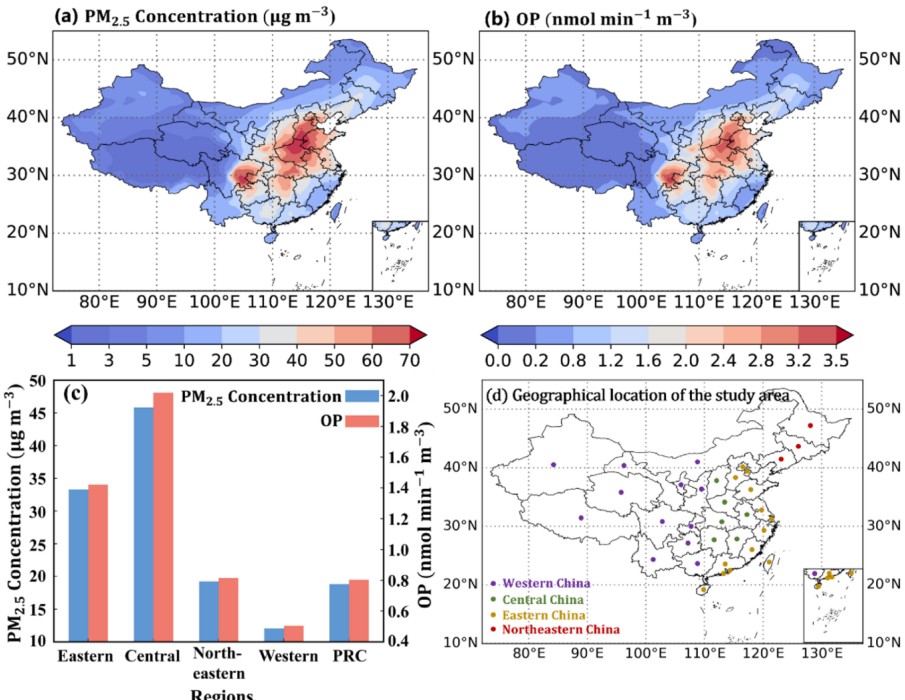

**Figure 6.** Spatial clustering of annual mean PM$_{2.5}$ concentrations (**a**) and annual mean OP (**b**) in China, annual mean PM$_{2.5}$ concentrations and annual mean OP (**c**) in different regions of China in 2014 under scenario C$_1$, and geographical location of the study area (**d**). The central region of China comprises Shanxi, Anhui, Jiangxi, Henan, Hubei, and Hunan provinces. The eastern region of China comprises Beijing, Tianjin, Hebei, Shanghai, Jiangsu, Zhejiang, Fujian, Shandong, Guangdong, Hainan, Hong Kong, Macao, and Taiwan. It should be noted that the eastern region in this study includes Hong Kong, Macao, and Taiwan. The western region of China consists of 12 provinces (autonomous regions and municipalities): Inner Mongolia, Guangxi, Chongqing, Sichuan, Guizhou, Yunnan, Tibet, Shaanxi, Gansu, Qinghai, Ningxia, and Xinjiang. The northeastern region of China comprises Liaoning, Jilin, and Heilongjiang provinces. Publisher's remark: please note that the above figure contains disputed territories.

**Table 2.** Goodness-of-fit test results for China in 2014 under scenario C$_1$.

| Item | Goodness-of-fit test | Gamma | Lognormal | Exponential |
| --- | --- | --- | --- | --- |
| PM$_{2.5}$ concentrations | SSE | **0.002** | 0.023 | 0.003 |
| | KS_pvalue | **0.329** | 0.000 | 0.000 |
| OP | Sumsquare_error | **0.654** | 0.746 | 1.209 |
| | KS_pvalue | 0.231 | **0.271** | 0.000 |

Bold values indicate the best results. * Note: SSE is the sum of squared error; KS_pvalue is the *P* value of the Kolmogorov–Smirnov test.

## 3.4 Contribution of anthropogenic emission sources to PM$_{2.5}$ and OP

To determine the impact of anthropogenic emissions on PM$_{2.5}$ and OP, we quantified their percent contribution (Fig. 10). Secondary aerosol formation, biomass burning, industrial, coal combustion for residential heating, and transportation sources contributed 48 %, 21 %, 21 %, 6 %, and 4 % to PM$_{2.5}$, respectively. Secondary aerosol formation, biomass burning, coal combustion for residential heating, industrial sources, and transportation sources contributed 58 %, 21 %, 11 %, 9 %, and 1 % to OP, respectively. This means that secondary aerosol formation and biomass burning are the main sources of PM$_{2.5}$ and OP.

Thus, we explored the spatial distribution characteristics of PM$_{2.5}$ and OP from different anthropogenic sources to reveal the spatial contributions of PM$_{2.5}$ concentrations and OP, as shown in Fig. 11. It was observed that the spatial distribution features of PM$_{2.5}$ concentrations and OP from each emission source are similar to those in Fig. 6, and they all adhere to the principle that the eastern region is higher than the western.

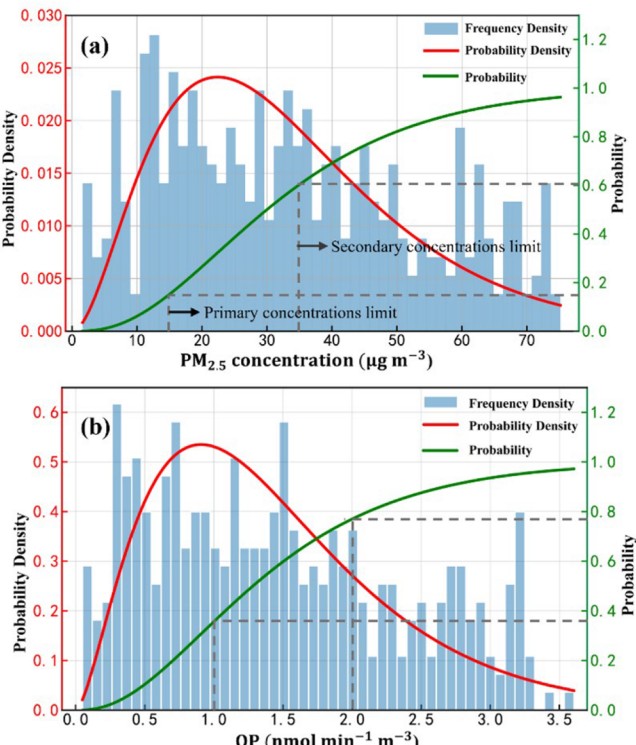

**Figure 7.** Probability distribution of **(a)** annual mean PM$_{2.5}$ concentrations and **(b)** annual mean OP for China in 2014 under scenario C$_1$.

It can be seen from Fig. 11 that the main reason that secondary aerosol formation is the main anthropogenic source of both PM$_{2.5}$ concentrations and OP in China is due to the higher pollution levels, more contributions to mass, and toxicity in the central and eastern regions. A relevant study (Molina et al., 2023) has highlighted the significant contribution of secondary aerosol formation to particle mass and intrinsic OP.

Chinese crops (especially corn straw), power plants mainly being concentrated in the central and eastern regions as well as the northeast and part of the western region, and the bigger intrinsic OP (Eq. 1) result in biomass burning becoming the second contribution.

In this study, coal combustion refers to coal heating from the residential sector. Coal burning increases secondary inorganic and organic aerosols in the air (Liu et al., 2018), which leads to stronger oxidative toxicity. Thus, due to greater heating demand in locations with a high population density and chilly winters (e.g., the northern part of central and eastern China), PM$_{2.5}$ concentrations and OP linked to coal burning are higher.

Industrial emissions are mainly derived from specific industrial processes in iron and steel, metallurgical production plants for non-ferrous metals (e.g., titanium and molybdenum), and so on. This is one main source for metals. Due to

the correlation between these transition metals and OP (Fang et al., 2017; Liu et al., 2018), China's four industrial zones (Liaozhong-South Heavy Industry Base, Beijing–Tianjin–Tangshan Industrial Base, Shanghai–Nanjing–Hangzhou Industrial Base, and Pearl River Delta Light Industry Base) are important contributors to PM$_{2.5}$ and OP emissions from industrial sources.

The transportation sector in the DEHM only considers tailpipe emissions, excluding non-exhaust emissions from vehicles like road dust, brake dust, and tire wear, which is a main reason that the traffic sources exhibit the lowest contribution to PM$_{2.5}$ concentrations and OP. Moreover, it can be observed from Fig. 11 that the sector's emissions are mainly concentrated in a small number of regions, such as Henan, Hebei, and Shandong. This is valid for the top three provinces in terms of vehicle particulate matter and nitrogen oxide emissions in 2014 according to the China Annual Vehicle Pollution Prevention and Control Report (MEE, 2015).

## 3.5 Uncertainty of OP estimates

OP is considered an important indicator of PM$_{2.5}$ toxicity and is associated with adverse health effects. Linking the predicted health effects of aerosols to OP may be more relevant than considering PM$_{2.5}$ mass alone (Alwadei et al., 2020). However, previous studies of OP in China have mainly focused on local areas, and OP and its sources are very different in space and time (Wen et al., 2023), which makes health research on OP challenging. At present, there are two kinds of methods for evaluating OP of PM$_{2.5}$: the cellular method and non-cellular method. The reproducibility of cellular methods is poor, and it is difficult to achieve a large sample size analysis. And the choice of cell type or cell line can significantly affect the OP results (Xing et al., 2023). The non-cellular method has the advantages of fast speed, simple operation, high reproducibility, and low cost. The most common non-cellular methods are DTT, AA, GSH, and 2′, 7′-dichlorofluorescein (DCFH) assays (Pietrogrande et al., 2019). However, standardized experimental methods for evaluating OP have not been established (Song et al., 2021), and it is difficult to provide more consistent data on OP across samples at different locations and times. Moreover, each non-cellular OP assay is specific for ROS, meaning that none of the methods are used as a standard method for assessing the toxicity of environmental particles. Several studies have used a multi-measure approach to compensate for the specificity of a single-probe response to ROS (Calas et al., 2018; Puthussery et al., 2020; Yu et al., 2021; Xu et al., 2021). Xu et al. (2021) used three measurement methods (OP$^{DTT}$, OP$^{AA}$, and OP$^{GSH}$) to estimate the Canadian annual mean OP and found that the sensitivity of the three methods to different components varied widely. Choosing a variety of methods for OP measurement can lead to more comprehensive results, but it can also lead to a significant increase in workload.

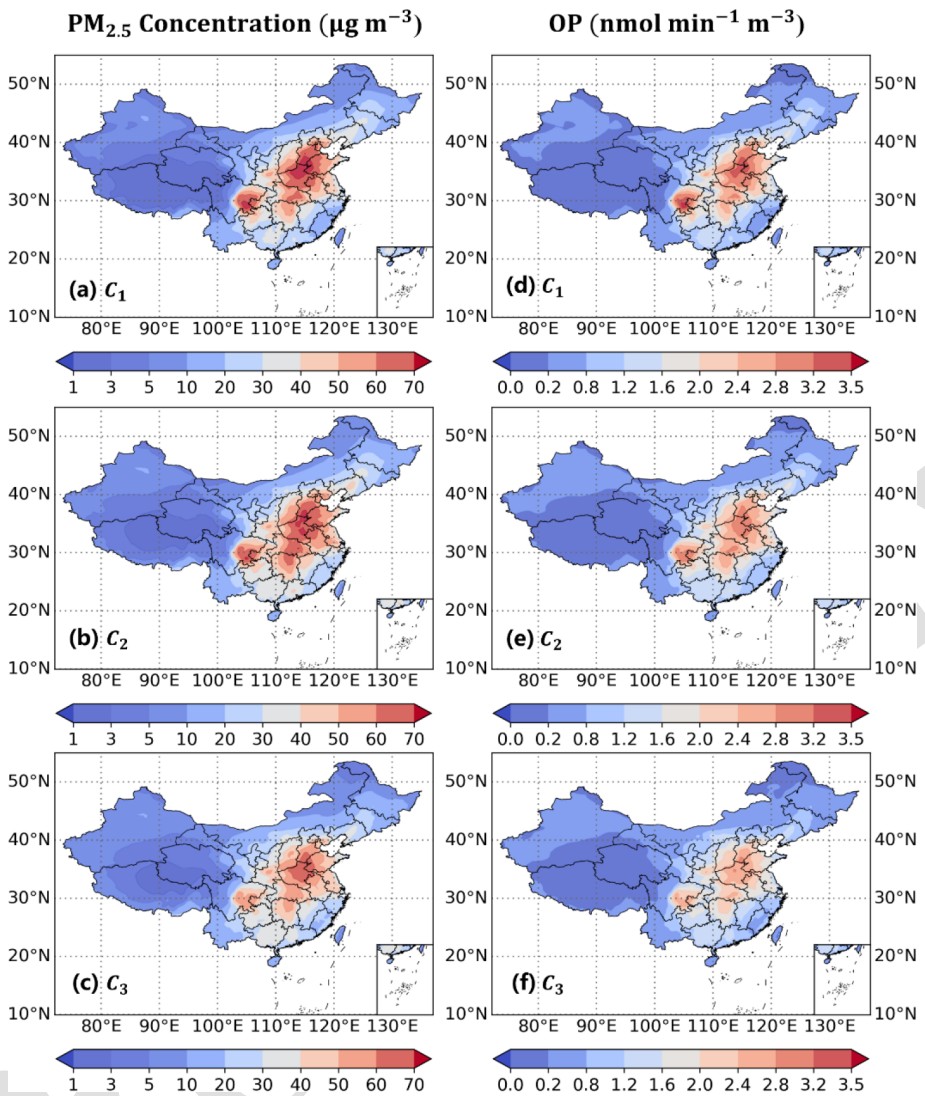

**Figure 8.** Spatial distribution of annual mean PM$_{2.5}$ concentrations and annual mean OP for China in 2014 under different scenarios. Panels **(a)**–**(c)** are PM$_{2.5}$ concentrations in scenarios $C_1$, $C_2$, and $C_3$, respectively. Panels **(d)**–**(f)** are the OP in scenarios $C_1$, $C_2$, and $C_3$, respectively. The meteorological datasets (emission inventories) employed for scenarios $C_1$, $C_2$, and $C_3$ are ERA5 (EDGAR-HTAP), ERA5 (Eclipse V6), and CESM (Eclipse V6), respectively. Publisher's remark: please note that the above figure contains disputed territories.

For this, we propose a hybrid approach combining existing observations of OP with a CTM. So, using OP from assays and their observed links to sources and chemical constituents can then be parameterized and implemented in a CTM for a comprehensive assessment of OP exposure over large areas and time periods. The method considers the seasonal characteristics of the chemical composition of PM$_{2.5}$ and the DTT activity measurement of PM$_{2.5}$. A positive matrix factorization (PMF) model and multiple linear regression (MLR) model were used to quantify the contribution of PM$_{2.5}$ emission sources to OP (the volume-normalized DTT activity, DTT$_v$). The normalized regression equation in this study provides the sensitivity of OP to each identified source. The advantage is that directly applying predicted and readily

available PM$_{2.5}$ data makes it easy to estimate the OP and assess health risks over large regions and across time and space. This approach enables the exploration of spatial and seasonal variations in aerosol OP across China, providing insight into the contribution of sources, atmospheric processes, and meteorological conditions. There are some main limitations to this study that may lead to uncertainty in predicting OP outcomes. Firstly, the OP prediction considered is incomplete and does not include all sources of OP. For example, the transportation sector refers only to road transport, excluding emissions from ships and other mobile sources. And the transportation sector only considers tailpipe emissions in traffic. These lead to some OP uncertainties. Additionally, this study only considers the intrinsic OP of total

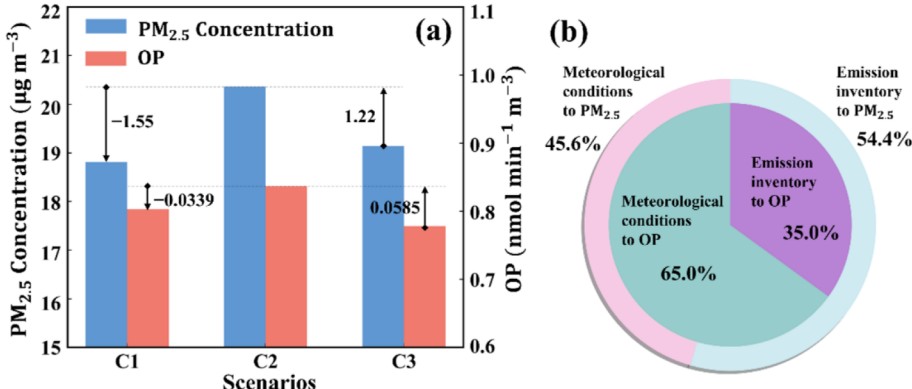

**Figure 9.** (a) Average annual PM$_{2.5}$ concentrations and average annual OP for China in 2014 under different scenarios. (b) The relative contribution of meteorological conditions and emission inventories to average annual PM$_{2.5}$ and average annual OP for China in 2014, with the outer circle representing PM$_{2.5}$ and the inner circle representing OP. The meteorological datasets (emission inventories) employed for scenarios C$_1$, C$_2$, and C$_3$ are ERA5 (EDGAR-HTAP), ERA5 (Eclipse V6), and CESM (Eclipse V6), respectively.

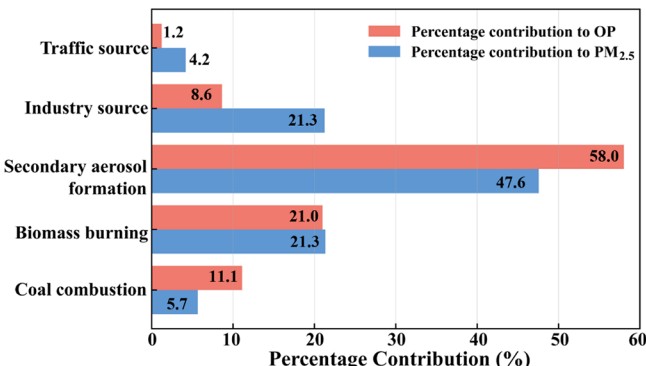

**Figure 10.** Percentage contribution of different anthropogenic sources (coal combustion for residential heating, biomass burning, secondary aerosol formation, industry, and traffic) to total PM$_{2.5}$ concentrations and OP for China in 2014 under scenario C$_1$.

SOA due to limited long-term measurements of SOA. Different types of SOA may exhibit varied OP responses due to differences in their sources, formation pathways, and chemical compositions. Aging and fresh SOA may also exhibit varying toxicities (F. Liu et al., 2023). In future research, efforts should be made to comprehensively collect PM$_{2.5}$ samples from various sources and fully explore the potential relationships between OP and PM$_{2.5}$ components and sources to further improve OP prediction models and reduce prediction uncertainties. Secondly, the OP prediction model adopted in this paper is based on Liu et al. (2018). The data samples are from DTT experimental measurements conducted in various coastal cities and from different emission sources, with limited data samples. In this study, they are used to predict the OP of cities across the country, which inevitably leads to a slight error in the forecast results. However, due to the spatiotemporal and emission source differences of the data samples were considered, the three cities selected are repre-

sentative, which reduces the errors caused by the data samples to a certain extent. Thirdly, a significant body of literature also indicates that vertical resolution can reflect atmospheric thermodynamic environments and the evolution processes of mesoscale systems, which are related to the diffusion and transport of PM$_{2.5}$. Insufficient vertical resolution can hinder the accurate prediction of PM$_{2.5}$ surface concentrations (Hara, 2011; Li et al., 2022; Li et al., 2023). Therefore, the configuration of the DEHM (e.g., vertical resolution) also introduces uncertainties to this study. In conclusion, the results calculated by the method proposed in this study are compared with existing measurement data (Liu et al., 2014; Liu et al., 2018; Wang et al., 2019), and good agreement is observed. For instance, through DTT measurements, Zhang et al. (2023) reported an average OP$_V^{DTT}$ of 1.33 nmol min$^{-1}$ m$^{-3}$ from January 2020 to June 2021 in downtown Nanjing, located in the Yangtze River Delta region of China, with a range of 0.82–2.08 nmol min$^{-1}$ m$^{-3}$. This is close to our estimated results (Liu et al., 2024) for the Yangtze River Delta region, where the annual mean OP during 2010–2014 was 1.56 nmol min$^{-1}$ m$^{-3}$, and the annual mean OP values for 2020 under two emission reduction scenarios were 1.36 nmol min$^{-1}$ m$^{-3}$ and 1.25 nmol min$^{-1}$ m$^{-3}$, respectively. The relative errors for the two scenarios were 2.3 % and 6.0 %. Another study (Liu et al., 2020) investigating the OP of PM$_{2.5}$ in Wuhan, located in the central China region, reported a mean OP$_V^{DTT}$ of 1.8 nmol min$^{-1}$ m$^{-3}$ for the summer of 2012 in downtown Wuhan. This aligns closely with our estimated results (Liu et al., 2024) for the central China region (1.73 nmol min$^{-1}$ m$^{-3}$), with a relative error of 3.9 %. Therefore, the method proposed in this study is reliable. The proposed method provides a possibility to solve the difficulty and high cost of OP measurement.

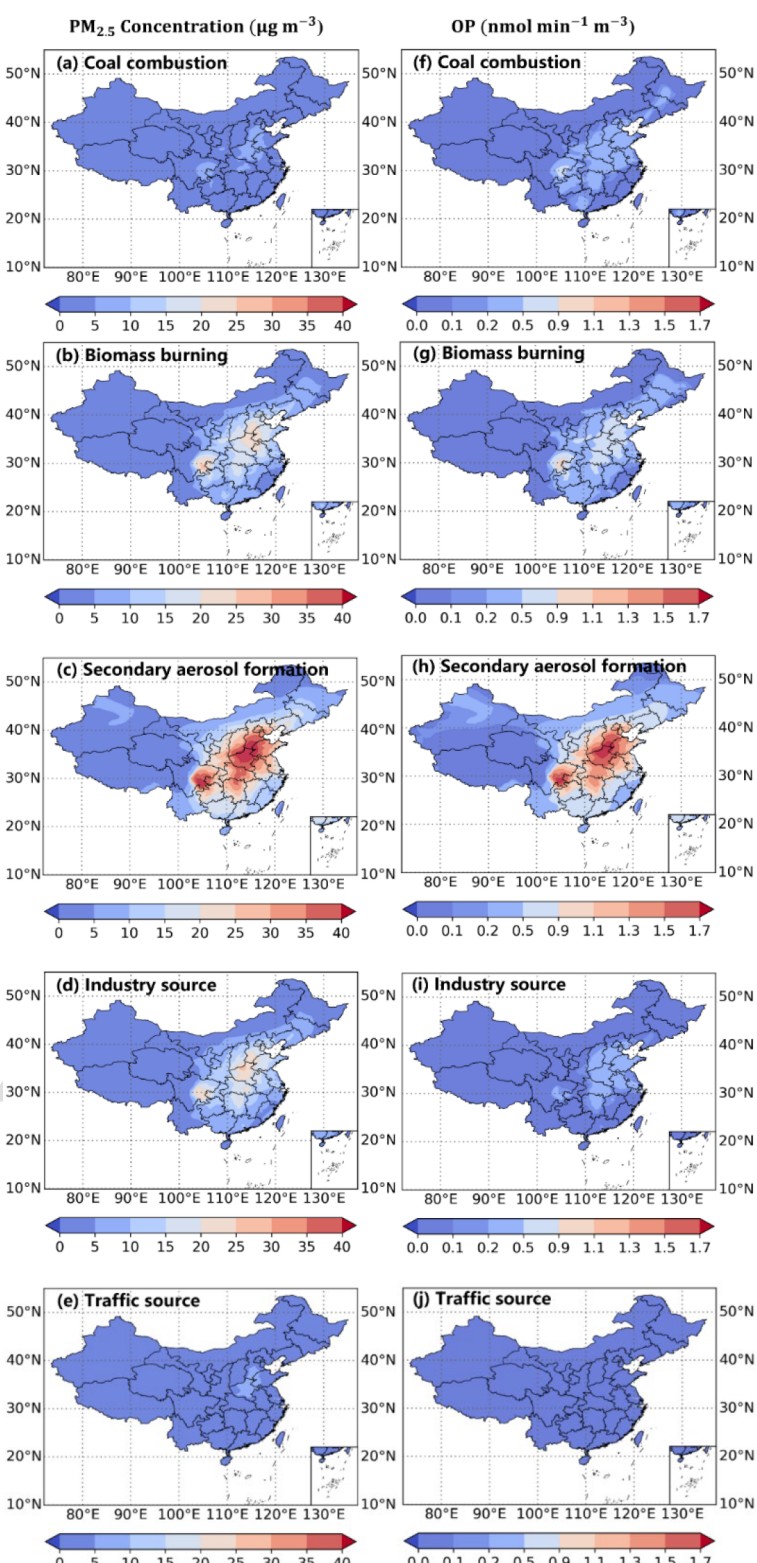

**Figure 11.** Spatial distribution of annual mean PM$_{2.5}$ concentrations and annual mean OP from different anthropogenic sources for China in 2014 under scenario C$_1$. Panels **(a)**–**(e)** are PM$_{2.5}$ concentrations derived from coal combustion for residential heating, biomass burning, secondary aerosol formation, industry, and traffic, respectively. Panels **(f)**–**(j)** are the OP derived from coal combustion for residential heating, biomass burning, secondary aerosol formation, industry, and traffic, respectively. Publisher's remark: please note that the above figure contains disputed territories.

## 4 Conclusions

This study established spatial modeling for PM$_{2.5}$ concentrations and OP, provided a method for calculating OP across China, and quantitatively assessed the impacts of meteorological conditions and anthropogenic emissions on PM$_{2.5}$ and OP variability and levels in China. The following conclusions can be drawn.

PM$_{2.5}$ and OP exhibited spatial clustering characteristics, with higher values mainly located in the central and eastern urban areas. About 85 % and 40 % of the areas had PM$_{2.5}$ annual average concentrations exceeding the primary concentration limit (15 µg m$^{-3}$) and secondary concentration limit (35 µg m$^{-3}$), respectively. Additionally, about 36 % of the areas had OP concentrations lower than 1 nmol min$^{-1}$ m$^{-3}$, while 23 % of the areas had OP concentrations higher than 2 nmol min$^{-1}$ m$^{-3}$.

Variability in both PM$_{2.5}$ and OP is influenced by a combination of meteorological conditions and emission inventories. Meteorological conditions contributed about 46 % of PM$_{2.5}$ variation and 65 % of OP variation. The emission inventory contributed about 54 % of the change in PM$_{2.5}$ and about 35 % of the change in OP.

The percentage contributions of secondary aerosol formation, biomass burning, industry, coal combustion for residential heating, and traffic to PM$_{2.5}$ were about 48 %, 21 %, 21 %, 6 %, and 4 %, respectively. The percentage contributions of secondary aerosol formation, biomass burning, coal combustion for residential heating, industry, and traffic to OP were approximately 58 %, 21 %, 11 %, 9 %, and 1 %, respectively.

A main finding of this study is that meteorological variability is the prime driver of OP variability, not emissions. Furthermore, secondary aerosol formation and biomass burning are the main sources of OP. Thus, air pollution strategies should focus more on biomass burning and the emissions of the precursors taking part in secondary aerosol formation, and it would be efficient to introduce special emissions controls during stagnation or other periods during which OP accumulates.

**Data availability.** Data from all DEHM simulations and post-processing codes are available from the corresponding author on request. The hourly observation data were obtained from the Ministry of Ecology and Environment of China (MEE, 2014, https://www.mee.gov.cn/).

**Supplement.** Density scatterplots of model performance and validation based on monthly mean observations (Fig. S1) are provided in the Supplement. The supplement related to this article is available online at: https://doi.org/10.5194/acp-24-1-2024-supplement.

**Author contributions.** JL performed the simulation and its validation and worked on data analysis, investigation, and writing (original draft). JHC conducted investigation, worked on methodology, and performed validation. ZY conducted the investigation, data analysis, and methodology. SD provided suggestions for data analysis and manuscript feedback. CG conducted the investigation and provided suggestions on the simulation. JB conducted the investigation. AN conducted the investigation and provided manuscript feedback. YY provided manuscript feedback, supervision, and funding. UI provided the resources, supervised this work, provided manuscript feedback, and managed the project administration. All authors discussed the results and commented on the manuscript.

**Competing interests.** The contact author has declared that none of the authors has any competing interests.

**Disclaimer.** Publisher's note: Copernicus Publications remains neutral with regard to jurisdictional claims made in the text, published maps, institutional affiliations, or any other geographical representation in this paper. While Copernicus Publications makes every effort to include appropriate place names, the final responsibility lies with the authors.

**Acknowledgements.** This study was supported by the National Natural Science Foundation of China (grant no. 52041601). The work of Liu Jiemei was also supported by the China Scholarship Council (CSC) under the State Scholarship Fund. Athanasios Nenes acknowledges support from the project PyroTRACH (ERC-2016-COG) (funded from H2020-EU.1.1. – Excellent Science – European Research Council – ERC, project ID 726165) and from the European Union project EASVOLEE funded from HORIZON-CL5-2022-D5-01 (project ID 101095457).

**Financial support.** This research has been supported by the National Natural Science Foundation of China (grant no. 52041601), the China Scholarship Council (CSC, grant no. 202206120146), the project PyroTRACH (ERC-2016-COG) (funded from H2020-EU.1.1. – Excellent Science – European Research Council – ERC, project ID 726165), and the European Union project EASVOLEE funded from HORIZON-CL5-2022-D5-01 (project ID 101095457).

**Review statement.** This paper was edited by Frank Keutsch and reviewed by two anonymous referees.

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

## Remarks from the language copy-editor

**CE1** Please note that the issue with the maps provided is the dashed line around disputed islands. The options are to remove this dashed line (and provide new maps without the dashed line) or to leave the disclaimer in the figure captions as it is (Publisher's note: this figure contains disputed territories). Our policy on this is not flexible, and one of the two options is required for publication. If you review papers published by Copernicus within the last few weeks/months, you can see that this updated policy has been consistently followed. Thank you in advance for your understanding.