# Peer review of "Impact of Meteorology and Aerosol Sources on PM2.5 and Oxidative Potential Variability and Levels in China"

_EGUsphere, 2023_

## Referee Comment (RC2)

Review of "Impact of Meteorology and Aerosol Sources on PM2.5 and Oxidative Potential Variability and Levels in China" by Liu et al.

The paper describes model calculations of PM2.5 and oxidative potential, which provides information on the impact of PM2.5 on human health. The paper examines the role of meteorology and anthropogenic emissions on PM2.5 concentrations and oxidative potential, finding that meteorological factors contribute more to the predicted surface PM2.5 concentrations and oxidative potential than anthropogenic emissions.

There have been many studies investigating PM2.5 distributions across China, but what is new here is taking those PM2.5 concentrations and estimating the oxidative potential. The results presented in this paper are not especially novel but give an incremental advancement in connecting air pollution dominated by particulate matter to human health. Like many papers, there is much to be clarified in the methodology and interpretation of results. However, a major need of the paper is a discussion of the uncertainties in the results and conclusions. In a sense, the authors address a couple of those uncertainties by performing simulations with different meteorology and different anthropogenic emissions. However, there is no discussion of the uncertainties associated with the assumptions made in their approach.

**Major Comments**

1. It would be good to see more explanation of what oxidative potential means. What does a value of 2 nmol/min/m3 imply? Oxidative potential is defined as "the ability of PM2.5 to produce reactive oxygen species (ROS) to *in-vitro* that consume intracellular antioxidants". The part in quotes is from line 41 of the paper and does not fully make sense to me. Is it meant that reactive oxygen species consume intracellular antioxidants as observed in lab studies? Does an oxidative potential mean that 2 nmol/m3 of ROS are produced per minute? Is that a lot? Are there thresholds for high oxidative potential versus low oxidative potential?

   I found a much clearer explanation of oxidative potential in Yang et al. (2021). I suggest revising the Introduction of this paper to present a clearer explanation, especially for readers who have not learned about oxidative potential yet.

2. The paper quantifies the annual average PM2.5 concentrations and oxidative potential for all of China and then discusses the spatial distribution of these parameters using maps. Distinct regions are noted in the spatial distribution discussion that I advocate should be quantified on a regional scale. What I mean by "quantify" is to provide average values of PM2.5 and OP for each region as was done in Figure 9a. The authors are the better experts to distinguish these regions, but what stands out in Figure 6 are these regions: northeast China, central and southeast China, southern portion of western China, and northwest China. As northwest China has the poorest agreement with observations, I suggest treating that region separately from the other regions in China.

3. The seasonality evaluation plots were interesting and point to good agreement during April – September and poorer agreement during the winter months. I suggest that the authors also separate their analysis between summer, when there is more confidence in the model results, and winter.

4. What role does PM2.5 from outside China have on PM2.5 concentrations and oxidation potential? For example, does biomass burning PM2.5 from Russia affect PM2.5 in China?

5. The study uses 50 km x 50 km horizontal grid spacing and 29 vertical levels to 100 hPa. What impact does this rather coarse resolution have on the results presented? For example, are the urban regions and their emissions well represented or do the emissions and concentrations get diluted by placing them in a grid box bigger than the urban region? Likewise, does the vertical resolution impact the prediction of the boundary layer height affecting the estimated surface concentrations of PM2.5? Please add a section discussing uncertainties in the study's results including the choice of the model configuration and other assumptions as noted below in the specific comments.

**Specific Science Comments**
1. Lines 28-29 of abstract: What is the meaning of OP values? Is 1 nmol/min/m3 low, and if so, does that mean there is small potential for health risk? Please explain why it is useful to report the numbers.

2. Section 2.2. It was not clear whether DEHM represents any feedbacks of aerosols on the meteorology. Could this be clearly explained in the model description.

3. Equation 1 (line 151) gives the calculation for oxidative potential as a function of PM2.5 concentrations. If I understand correctly, this equation comes from Liu et al. (2018) cited in the paper and is based on positive matrix factorization performed on samples from coastal cities. Could an explanation be added stating 1) units of the terms, and 2) the reliability of using this equation outside of an industrial coastal region. To me, this is another uncertainty embedded in this study.

4. Line 207. Why is the emissions reduction 30% and not another number? Please justify. Have any tests been done to learn about the response to different assumed emission reduction values? Perhaps this could be included in a section discussing uncertainties of the study's results.

5. Line 207. Could the authors clarify whether only PM2.5 emissions are reduced or if any precursors (e.g. SO2) also have emission reductions. If I understand the methods correctly, it appears that only primary PM2.5 emissions are perturbed for each sector. Since about half the aerosol is produced by chemistry (line 227) and emission controls can be placed on PM2.5 precursors (e.g., SO2 and NOx), it seems that additional calculations for reducing precursor species from different sectors is warranted. Could the authors please justify why only primary PM2.5 anthropogenic emission reductions are investigated, or add results presenting the impact of reducing PM2.5 precursors on the oxidative potential is examined.

6. Lines 218-225. Could information on the fraction of each sub-sector's contribution to E$re$ and E$pp$ be added?

7. Equation 9. Could the authors explain the 30% term in this equation? If it is the percentage reduction, then wouldn't this term change depending on which source sector is perturbed?

8. Figure 3. Are the results shown for simulation C1? Why not also show the evaluation for simulations C2 and C3? Throughout the manuscript, it would be informative to see the figures for all three scenarios. Please add these figures in either the main manuscript or the supplement.

9. Are the Figure 4 results for the annual mean? If so, please state this and explain if it is subtracting the annual mean of the simulation results from the annual mean of the observations. In contrast, it could be differencing individual time points for each location and then taking an annual mean of the difference.

10. Figure 4 shows differences of up to 18 ug/m3 in central China. How does this compare to the annual mean concentrations? It would be helpful to also see a percent difference map.

11. Line 297. I disagree with the statement, "We conclude that the model performs well ….", which is stated just after noting the poor agreement in northwest China (which is for good reasons). Please qualify this statement. I suggest limiting the remainder of the analysis to all of China except northwest China. See also my comments about performing calculations for specified regions.

12. Lines 309-320. Do the maps of PM2.5 concentration and OP reflect the anthropogenic emissions map of PM2.5? As written, this text does not tell me anything new, but I think the authors were hinting at some useful information in the last few lines. I suggest adding maps of anthropogenic emissions for each sector to support why we see the spatial distribution in Figure 6. Then rewrite these lines to focus on these connections.

13. Section 3.4 provides good conclusions and interesting points but does not complete the analysis of their model output to explain the results. Instead of going into detail on each panel shown in Figure 10, I suggest highlighting what is learned. What I learned is that secondary aerosol formation is the primary contributor to PM2.5, while biomass burning and industrial sources each contribute moderately to PM2.5 and residential and traffic emissions are small (negligible?) contributions. I also learned that residential burning has more of an impact on the oxidative potential because it is weighted more (based on equation1) than the other terms. What I did not learn is why secondary aerosol formation is the largest contributor to PM2.5 and oxidative potential. Are precursor emissions much greater than PM2.5 emissions? What role do oxidants play in controlling secondary aerosol formation? As a consequence of this result, should studies on source attribution to oxidative potential focus on precursor emissions and their source sectors? I did not see any DEHM results focused on carbonaceous aerosols or metals and the oxidative potential equation does not make use of that information. Although these aerosols are key culprits affecting human health, I do not understand

why the text about biomass combustion, coal burning, and traffic emissions discuss these details without supporting information from the DEHM results.

**Organization, Clarity, Technical Comments**

1. Lines 29-31. Please write this sentence more clearly: meteorological conditions contributed 46% and 65% to PM2.5 concentrations and oxidative potential, respectively, while anthropogenic emissions contributed 54% and 35% to PM2.5 concentrations and oxidative potential, respectively.

2. Line 87. "combing" should be "combining".

3. Line 105. Please explain further how the DTT measurements are brought into this study. Do you mean via the positive matrix factorization done in a previous study?

4. Section 2. I think it would be good to reorganize the section. Currently section 2.3 interrupts the discussion of model simulations in 2.2 and 2.4. Perhaps the explanation of oxidative potential could be presented first followed by the control and sensitivity simulation configurations.

5. Section 2.2, meteorology setup. Please give more detail. I assume the nested domain at 50 km x 50 km horizontal grid spacing was used for the analysis, but this was not stated explicitly. I also assume that the outer domain provided initial and boundary conditions for the nested domain, but it was not stated. Was there a spin-up period for the simulations before conducting the analysis for 2014? How frequently was the meteorology data updated to reanalysis (or nudged to reanalysis)? Why was ERA5 chosen to drive the WRF model and not another global reanalysis product like MERRA or NCEP?

6. Section 2.2, chemistry configuration. It would be useful to give a description of the gas-phase chemistry and how PM2.5 is formed. For example, what hydrocarbons are included that would contribute to SOA formation? What sulfur chemistry and nitrogen oxides chemistry are represented that make sulfate and nitrate aerosol? Consider including the list of chemical reactions in the supplement. Are the aerosols represented with a bulk aerosol scheme?

7. Section 2.2, emissions. I suggest giving short descriptions of each emissions source: How are biogenic emissions, sea salt emission, lightning emissions calculated and what are they emitting? Is biomass burning from wildfires included? For anthropogenic emissions, it states EDGAR-HTAP is used, but it does not include a description of what that inventory emits and what grid spacing the inventory has. Yet, in section 2.4 there is a paragraph giving that information for the ECLIPSE emissions. It would be good to have similar information about each inventory so that the reader can better understand why there may be differences between simulation C1 and simulation C2. Especially useful would be to report the emission inventories' annual values for China for PM2.5 and key precursors (e.g. SO2) as this will provide quantitative information on how EDGAR-HTAP and Eclipse differ.

8. Line 152. Please add *se* and its definition.

9. Line 155. "Industry source is primarily from specific industry processes" is not providing any insight as to what kind of industry or specific types of emissions. Please give more information.

10. Section 2.4.1. There are a number of reanalysis datasets available (e.g., ERA5, MERRA, NCEP FNL). Why were ERA5 and CESM chosen?

11. Line 167. "reanalyse" should be "reanalysis"

12. Line 172. "was first ran" should be "was first run"

13. Line 195. I suggest putting the sentence, "abs represents the absolute value" at the end of the paragraph, and rewrite to something more readable, e.g. "In the equations, the abs function represents the absolute value of the quantity in parentheses."

14. Line 203. I suggest using "described" instead of "proposed".

15. Line 225. Change "More and more" to "Previous".

16. Line 257. Please maintain the same verb tense. I suggest "are mainly … are limited".

17. Line 263. Change "were" to "are".

18. Section 3.1. Please specify which model domain is being evaluated.

19. Line 275. Is it MME or MEE?

20. Diff_si-ob is used to express the difference between simulated and observed values. The way this term is written implies that it equals the simulation value minus the observations value. However, the values in Figure 4 appear to be observations minus simulated values. Could the authors clean up the terminology please.

21. Line 293. Change "are" to "were.

22. Line 293-294. Remove "with Figure S1a … December".

23. Line 306-307. First sentence needs to be written better to something like: To learn about the spatial distributions of PM2.5 concentrations and OP, we plot maps of surface PM2.5 and OP for scenario C1 (Figure 6).

24. Line 308. The sentence, "The findings … and OP" is not needed.

25. Line 313. Is the term "urban areas" for low OP meant? Or is this area more rural?

26. Line 318. "northern residents in China right region" does not make sense to me.

27. Lines 336-340. I do not think so many significant digits are needed. I suggest using 85% instead of 84.8%, and likewise for the other numbers used here.

28. Line 349. Change to "illustrates".   Line 351. Change to "presents".

29. Line 350-351. The sentence is not needed as it repeats the figure caption.

30. Line 392. Change "are" to "is".

31. Line 430 and line 432. I recommend reducing the number of significant digits.

**Figures and Tables**

1. In all figure captions that show results, please include information on the time period shown (e.g., annual average) and spatial region shown (where appropriate).

2. Figure 3, figure caption. Please state what parameter is being plotted. I assume PM2.5, but it should be explicitly stated.

3. Figure 3 would benefit from having less white space. I suggest changing the maximum value to 120 or 150 ug/m3.

4. Figure 3. Are the points shown in panel b for the Dalhousie dataset for the same locations as the MEE observations? Or are there more points taking advantage of the gridded dataset?

5. Figure 4. Adjust the colorbar so that the whitest color is zero. That makes it easier to see differences between positive and negative values.

6. Figure 5. Are the observations shown in the figure from the MEE data or the Dalhousie reanalysis? Please note this in the figure caption.

7. Figure 5. It is difficult to discern the dashed and solid horizontal lines because the solid line does not extend from one edge of the colored region to the other. Is it possible to fix this?

8. Figure 8. Using 2 rows and 3 columns makes for smaller panels. I suggest using 3 rows and 2 columns (transposing the panels). I also suggest adding titles for each column, "PM2.5 (units)" and "OP (units)" and then panel labels that simply are the simulation name.

9. Figure 8. What are the insets showing in the bottom right of each panel? They are not discussed, so I suggest removing them.

10. Figure 10. Like Figure 8, I suggest using 5 rows and 2 columns instead of 4 rows and 3 columns. I also suggest adding titles for each column, "PM2.5 (units)" and "OP (units)" and then panel labels are the sector source (e.g., residential heating).

11. Figure 11. A more complete figure caption is needed: Percent contribution of different anthropogenic sources (traffic, industry, secondary aerosol formation, biomass burning, coal combustion) to total PM2.5 concentration and oxidation potential.

**References**

Yang, F., C. Liu, H. Qian, Comparison of indoor and outdoor oxidative potential of PM2.5: pollution levels, temporal patterns, and key constituents, *Environment International*, **155**, 2021, https://doi.org/10.1016/j.envint.2021.106684.

---

## Author Comment (AC1)

Manuscript: egusphere-2023-2615

Title: Impact of Meteorology and Aerosol Sources on PM$_{2.5}$ and Oxidative Potential Variability and Levels in China

We appreciate all the valuable comments from the reviewers and editor, which significantly improved the quality of the manuscript. We have studied the comments carefully and revised the manuscript accordingly. The comments and our responses point-to-point are listed below. Changes to the paper are shown in blue, so that the reviewers can easily review them again.

Replies to Reviewers and Editor

First of all, we thank both reviewers and editor for their positive and constructive comments and suggestions.

Reviewers' comments:

RC1:

In this article Liu et al. use the Danish Eulerian Hemispheric Model (DEHM) to evaluate the oxidative potential (OP) of particulate matter as a function of meteorology and PM source on of aerosols in China.

The author's make use of the parametrization of OP published in 2018 by Liu W. et al. This parametrization is quite simplistic:

DTTv = 0.088 × Coal combustion+0.076 × Biomass burning+0.041 × Secondary source+0.034 × Industry+0.017 × Traffic source (coefficient with a unit of nmol·min−1·μg source−1)

Which is quite convenient for implementing in models.

Using this approach the authors evaluate the importance of different PM sources as well as weather on OP.

The author's approach is valid based on their assumptions. However, it hinges on this parametrization being valid for the entirety of China which may not be the case. In addition, it is known that different SOA subtypes can have varied OP responses (See Liu et al. Environ. Sci. Technol. 2023, 57, 38, 14150–14161) which the authors make no mention of or address in any capacity. This is important considering that SOA is the main source (58%) of OP in this study. While this parametrization cannot possibly represent the different types of SOA and their respective OP, it should be acknowledged in the text as a downside and a target for future measurement campaigns and modeling efforts.

Although these downsides should be discussed in the main article, they do not take away from the results of this work which I believe are novel and useful.

As for technical aspects, the authors describe their methods in a complete manner, clearly cite relevant sources, and use an appropriate title and abstract. The paper is well structured and easy to read and most of the data necessary for interpretation is included in the main text. It can be published after a discussion of the weaknesses of the implemented parametrization.

**Response**: Thank you very much for your acknowledgement and valuable comments. As you mentioned, the parameterization of OP may have different responses across different regions of China and various subtypes of

SOA. A comprehensive discussion of these aspects in the main text would enhance the completeness of the current study and facilitate further advancements in OP research in the future. Following your advice, we have added a section titled "Uncertainty of OP estimates" in the main text, which elaborates the limitations and uncertainty of the OP estimation method proposed in this study in detail (**Please see Pages 24-26**).

The detailed revisions in the manuscript are shown below:

[revised manuscript text omitted]

The paper describes model calculations of PM$_{2.5}$ and oxidative potential, which provides information on the impact of PM$_{2.5}$ on human health. The paper examines the role of meteorology and anthropogenic emissions on PM$_{2.5}$ concentrations and oxidative potential,finding that meteorological factors contribute more to the predicted surface PM$_{2.5}$ concentrations and oxidative potential than anthropogenic emissions.

There have been many studies investigating PM$_{2.5}$ distributions across China, but what is newhere is taking those PM$_{2.5}$ concentrations and estimating the oxidative potential. The results presented in this paper are not especially novel but give an incremental advancement in connecting air pollution dominated by particulate matter to human health. Like many papers, there is much to be clarified in the methodology and interpretation of results. However, a major need of the paper is a discussion of the uncertainties in the results and conclusions. In asense, the authors address a couple of those uncertainties by performing simulations with different meteorology and different anthropogenic emissions. However, there is no discussionof the uncertainties associated with the assumptions made in their approach.

**Major Comments**

1. It would be good to see more explanation of what oxidative potential means. What doesa value of 2 nmol/min/m3 imply? Oxidative potential is defined as "the ability of PM$_{2.5}$ to produce reactive oxygen species (ROS) to *in-vitro* that consume intracellular antioxidants". The part in quotes is from line 41 of the paper and does not fully make sense to me. Is it meant that reactive oxygen species consume intracellular antioxidants as observed in lab studies? Does an oxidative potential mean that 2 nmol/m3 of ROS areproduced per minute? Is that a lot? Are there thresholds for high oxidative potential versus low oxidative potential?

   I found a much clearer explanation of oxidative potential in Yang et al. (2021). I suggestrevising the Introduction of this paper to present a clearer explanation, especially for readers who have not learned about oxidative potential yet.

   **Response**: Thank you very much for your constructive comments. The comments have been answered in detail in the following order:

   **First,** the ambient PM$_{2.5}$ OP can be evaluated by the determination of dithiothreitol (DTT) consumption rates. Oxidative potential of PM$_{2.5}$ was detected by DTT assay, which estimates the redox activity of PM$_{2.5}$ by detecting the ability of the particulate redox active compounds to catalyze the transfer of electrons from DTT to oxygen. Hence, the DTT consumption rate is proportional to the concentration of redox active compounds in the PM$_{2.5}$ extracts. Volume-normalized OP (OP$_V$) was the rate of DTT consumed per minute per volume of air ($nmol\ min^{-1}\ m^{-3}$). Following your suggestion, we have provided a clearer explanation in the Introduction (**Please see Page 3**).

   **Second,** the value in $nmol\ min^{-1}\ m^{-3}$ represents the rate at which DTT is consumed per minute per volume of air. This study characterizes the distribution characteristics of OP across different regions of China using probability density functions. It was found that 35% of regions in China had an OP below 1 $nmol\ min^{-1}\ m^{-3}$, 41% of regions had an OP between 1 $nmol\ min^{-1}\ m^{-3}$ and 2 $nmol\ min^{-1}\ m^{-3}$, and 23%

had an OP higher than 2.00 $nmol\ min^{-1}\ m^{-3}$. This indicates that in 23% of regions in China, the rate of DTT consumption per volume of air exceeds 2.00 $nmol$. Liu et al. (2020) summarized OP measurements conducted in nine regions of China around 2014. The results showed that the average OP content in northern Beijing was highest during the winter of 2016 (~14.0 $nmol\ min^{-1}\ m^{-3}$), while the average OP level in Shanghai during the spring of 2016 was lowest (~0.15 $nmol\ min^{-1}\ m^{-3}$). Therefore, an OP of 2.00 $nmol\ min^{-1}\ m^{-3}$ may not be considered high. Currently, there are no reported thresholds for high or low OP levels. Similar to PM$_{2.5}$ concentrations, there may exist a threshold for OP, beyond which exposure could pose a threat to human health. Exploring the OP threshold will be our future endeavor.

The detailed revisions in the manuscript are shown below:

**Introduction (Page 3):**

"Many recent studies have suggested that the oxidative potential (OP) of PM$_{2.5}$ may better explain the negative impact of PM$_{2.5}$ exposure on human health than the well-established metric of mass concentrations (Yu et al., 2019; Gao et al., 2020). OP refers to the ability of PM$_{2.5}$ to induce oxidative stress (OS) (Yang et al., 2021)."

Gao, D., Pollitt, K., Mulholland, J.A., Russell, A.G., Weber, R.J. (2020). Characterization and comparison of PM2.5 oxidative potential assessed by two acellular assays. *Atmos Chem Phys*, 20, 5197-5210. https://doi.org/10.5194/acp-20-5197-2020.

Yang, F., Liu, C., Qian, H. (2021). Comparison of indoor and outdoor oxidative potential of PM2.5: pollution levels, temporal patterns, and key constituents. *Environ Int*, 155, 106684. https://doi.org/10.1016/j.envint.2021.106684.

Yu, S., Liu, W., Xu, Y., Yi, K., Zhou, M., Tao, S., Liu, W. (2019). Characteristics and oxidative potential of atmospheric PM2.5 in Beijing: Source apportionment and seasonal variation. *Sci Total Environ*, 650, 277-287. https://doi.org/10.1016/j.scitotenv.2018.09.021.

Liu, Q., Lu, Z., Xiong, Y., Huang, F., Zhou, J., Schauer, J.J. (2020). Oxidative potential of ambient PM2.5 in Wuhan and its comparisons with eight areas of China. *Sci Total Environ*, 701, 134844. https://doi.org/10.1016/j.scitotenv.2019.134844.

2. The paper quantifies the annual average PM$_{2.5}$ concentrations and oxidative potentialfor all of China and then discusses the spatial distribution of these parameters using maps. Distinct regions are noted in the spatial distribution discussion that I advocate should be quantified on a regional scale. What I mean by "quantify" is to provide average values of PM$_{2.5}$ and OP for each region as was done in Figure 9a. The authorsare the better experts to distinguish these regions, but what stands out in Figure 6 are these regions: northeast China, central and southeast China, southern portion of western China, and northwest China. As northwest China has the poorest agreement with observations, I suggest treating that region separately from the other regions in China.

**Response**: Thank you very much for your constructive comments. As you mentioned, this study discussed the spatial distribution of PM$_{2.5}$ and OP using maps. The study divided China into four economic regions: Northeast (including Liaoning, Jilin, and Heilongjiang provinces), Central (including Shanxi, Anhui, Jiangxi, Henan, Hubei, and Hunan provinces), East (including Beijing, Tianjin, Hebei, Shanghai, Jiangsu, Zhejiang, Fujian, Shandong, Guangdong, Hainan, Hong Kong, Macau, and Taiwan; it is noteworthy that the eastern region in this study includes the regions of Hong Kong, Macau, and Taiwan), and West (including Inner Mongolia, Guangxi, Chongqing, Sichuan, Guizhou, Yunnan, Tibet, Shaanxi, Gansu, Qinghai, Ningxia, and Xinjiang Autonomous Region, and municipalities directly under the central government). These four regions were evaluated in the model, revealing falling within an acceptable simulated bias in most areas of Northeast, Central, East China, and Western regions. Although it may be beneficial to separate the Northwestern region of China from the others, considering the limited availability of monitoring station data in the Western region,

even dividing the Western region into Southwest and Northwest regions may not substantially improve the evaluation. Furthermore, the Western region has lower population density and anthropogenic emissions compared to the other three regions, resulting in less pronounced impacts on PM$_{2.5}$ and OP. Therefore, the bias resulting from this discrepancy can be deemed acceptable. Overall, this study considers the division into Northeast, Central, East, and West regions as reasonable. To further characterize the spatial distribution, quantification of the spatial distribution on a regional scale, as you suggested, is necessary. Therefore, this study quantifies PM$_{2.5}$ and OP for four regions, as shown in Figure 6c (**Please see Pages 16-17**).

The detailed revisions in the manuscript are shown below:

**Results and discussion:**

"To learn about the spatial distributions of PM$_{2.5}$ concentrations and OP, we plot maps of surface PM$_{2.5}$ and OP for scenario C$_1$ (Figure 6a and 6b) and quantified the average annual PM$_{2.5}$ concentrations and OP across different regions of China (Figure 6c). Figure 6d depicted the geographical location of the study area. High PM$_{2.5}$ concentrations and High OP are mainly located in central and eastern urban clusters. Low PM$_{2.5}$ concentrations and Low OP are mainly distributed in northeastern and western China. The results in Figure 6c indicated that the annual average PM$_{2.5}$ concentrations/OP in eastern, central, northeastern, and western China are 33 $\mu g\ m^{-3}$ /1.4 $nmol\ min^{-1}\ m^{-3}$, 46 $\mu g\ m^{-3}$ /2.0 $nmol\ min^{-1}\ m^{-3}$, 19 $\mu g\ m^{-3}$ /0.8 $nmol\ min^{-1}\ m^{-3}$, and 12 $\mu g\ m^{-3}$/0.5 $nmol\ min^{-1}\ m^{-3}$, respectively." **(Page 16)**

"

[Figure]

Figure 6. Spatial clustering of annual mean PM$_{2.5}$ concentrations (a) and annual mean OP (b) in China, annual mean PM$_{2.5}$ concentrations and annual mean OP (c) in different regions of China in 2014 under scenario C$_1$, and geographical location of the study area (d); the central region of China comprises Shanxi, Anhui, Jiangxi, Henan, Hubei, and Hunan provinces; the eastern region of China comprises Beijing, Tianjin, Hebei, Shanghai,

Jiangsu, Zhejiang, Fujian, Shandong, Guangdong, Hainan, Hong Kong, Macao, and Taiwan; It should be noted that the eastern region in this study includes Hong Kong, Macao and Taiwan; the western region of China consists of twelve provinces (autonomous regions and municipalities): Inner Mongolia, Guangxi, Chongqing, Sichuan, Guizhou, Yunnan, Tibet, Shaanxi, Gansu, Qinghai, Ningxia and Xinjiang; the northeastern region of China comprises Liaoning, Jilin and Heilongjiang provinces." **(Page 17)**

3. The seasonality evaluation plots were interesting and point to good agreement during April – September and poorer agreement during the winter months. I suggest that the authors also separate their analysis between summer, when there is more confidence inthe model results, and winter.

**Response**: Thank you very much for your constructive comments. In the model evaluation, we discussed the model performance from both spatial and temporal perspectives to comprehensively assess DEHM's ability to simulate PM$_{2.5}$ in China. While seasonal discussions could provide clearer assessments of PM$_{2.5}$ and OP levels in each season, this study focuses on the annual contributions of meteorological conditions and anthropogenic emissions to PM$_{2.5}$ concentrations and OP, aiming for an overall assessment over the entire year. Therefore, evaluating the model based on annual average data provides a good overall assessment. Building upon this, we conducted a quantitative assessment of the spatial distribution characteristics and driving factors based on annual average data, exploring the spatial distribution of PM$_{2.5}$ and OP in China, as well as the impacts of meteorological conditions and anthropogenic emissions on both. The monthly and seasonal distributions of PM$_{2.5}$ and OP that you mention are also the focus of our other study, which we will expand on in more detail there.

4. What role does PM$_{2.5}$ from outside China have on PM$_{2.5}$ concentrations and oxidationpotential? For example, does biomass burning PM$_{2.5}$ from Russia affect PM$_{2.5}$ in China?

**Response**: Thank you very much for your constructive comments. On the issue of the impact of outside China PM$_{2.5}$ on China's air quality, existing studies (Xu et al., 2023) have utilized the GEOS-Chem chemical transport model to investigate the influence of foreign anthropogenic emissions (from Asian countries such as Bangladesh, Indonesia, India, Japan, South Korea, Malaysia, Myanmar, the Philippines, Thailand, and Vietnam) on PM$_{2.5}$ pollution in China in 2015. They found that, nationwide, foreign anthropogenic emissions in 2015 contributed approximately 2.4 $\mu g\ m^{-3}$ to PM$_{2.5}$ in China, accounting for 6.2% of the national average PM$_{2.5}$ concentration. Specifically, the contribution of foreign anthropogenic emissions was highest in the Eastern region of China (including Anhui, Hebei, Henan, Jiangsu, Liaoning, Shandong, Beijing, and Tianjin), with 5.0 $\mu g\ m^{-3}$ of PM$_{2.5}$ (8%); prominent cross-border pollution was also observed along the southwestern border of China, particularly affecting Yunnan Province with 4.9 $\mu g\ m^{-3}$ of PM$_{2.5}$ (18%), primarily originating from South Asia (i.e., India). Thus, outside China emissions have a certain impact on PM$_{2.5}$ concentrations in China. Therefore, in this study, the influence of transboundary transport on Chinese PM$_{2.5}$ was considered by setting up a research domain with the Northern Hemisphere as the parent domain and China as the nested domain, better representing the transport of pollutants from the Northern Hemisphere to China. However, it is challenging to isolate the impact of transboundary transport on PM$_{2.5}$ pollutions in China, and Xu et al. (2023) 's study revealed a relatively minor impact of transboundary transport on PM$_{2.5}$ pollutions in China. Therefore, this work did not separately quantify the influence of transboundary transport and focused on exploring the contributions of meteorological conditions and Chinese anthropogenic emissions to PM$_{2.5}$ pollutions.

Xu, J.W., Lin, J., Luo, G., Adeniran, J., Kong, H. (2023). Foreign emissions exacerbate PM2.5 pollution in China through nitrate chemistry. Atmos Chem Phys, 23, 4149-4163. https://doi.org/10.5194/acp-23-

4149-2023.

Sun, L., Yang, L., Wang, D., Zhang, T. (2023). Influence of the Long-Range Transport of Siberian Biomass Burnings on Air Quality in Northeast China in June 2017. Sensors (Basel), 23. https://doi.org/10.3390/s23020682.

5.  The study uses 50 km × 50 km horizontal grid spacing and 29 vertical levels to 100 hPa. What impact does this rather coarse resolution have on the results presented? For example, are the urban regions and their emissions well represented or do the emissions and concentrations get diluted by placing them in a grid box bigger than the urban region? Likewise, does the vertical resolution impact the prediction of the boundary layer height affecting the estimated surface concentrations of PM$_{2.5}$? Pleaseadd a section discussing uncertainties in the study's results including the choice of the model configuration and other assumptions as noted below in the specific comments.

**Response**: Thank you very much for your constructive comments. The horizontal resolution of 50 km x 50 km used in the nested domain of this study can be considered appropriate. Firstly, previous studies on the distribution of air pollutants and meteorological parameters in urban areas of China have utilized resolutions of 50 km × 50 km (Dai et al., 2021; Wang et al., 2013) or coarser (Gao et al., 2018), with their results indicating that this resolution is sufficient for simulating the distribution of air pollutants and meteorological parameters. Secondly, the validation results of the PM$_{2.5}$ prediction model in this study also support this point. Although the regions defined in this study (such as Eastern, Central, Western, and Northeast China) are smaller compared to the entire country, each region still encompasses numerous model grid cells, allowing for the representation of specific characteristics of each area. Therefore, the horizontal resolution of 50 km × 50 km adopted in this study is deemed appropriate. Additionally, vertical resolution affects the prediction of boundary layer height, and a significant body of literature also indicates that vertical resolution can reflect atmospheric thermodynamic environments and the evolution processes of mesoscale systems, which are related to the diffusion and transport of PM$_{2.5}$. Insufficient vertical resolution can hinder the accurate prediction of PM$_{2.5}$ surface concentrations (Zhang et al., 2023). Considering this aspect and in line with existing literature (MEE, 2013; Gao et al., 2018), this study configured 29 uneven layers for vertical distribution, with the highest layer reaching 100 hPa and the lowest layer at approximately 20 meters in height. The model evaluation results in Section 3.1 demonstrate that the model and its associated configurations can still predict PM$_{2.5}$ levels in various regions of China effectively. To further enhance the rigor of the study results, this study has incorporated discussions on the uncertainties related to model configurations and vertical resolution, as per your suggestion (**Please see Pages 24-26**).

Dai, T., Cheng, Y., Goto, D., Li, Y., Tang, X., Shi, G., Nakajima, T. (2021). Revealing the sulfur dioxide emission reductions in China by assimilating surface observations in WRF-Chem. *Atmos Chem Phys*, 21, 4357-4379. https://doi.org/10.5194/acp-21-4357-2021.

Gao, M., Beig, G., Song, S., Zhang, H., Hu, J., Ying, Q., Liang, F., Liu, Y., Wang, H., Lu, X., Zhu, T., Carmichael, G.R., Nielsen, C.P., McElroy, M.B. (2018). The impact of power generation emissions on ambient PM2.5 pollution and human health in China and India. *Environ Int*, 121, 250-259. https://doi.org/10.1016/j.envint.2018.09.015.

MEE. (2013). Technical guide for source analysis of atmospheric particulate matter. https://www.mee.gov.cn/gkml/hbb/bwj/201308/W020130820340683623095.pdf. Accessed 11 January, 2023.

Wang, S., Yu, E. (2013). Simulation and projection of changes in rainy season precipitation over China using the WRF model. *Acta Meteorologica Sinica*, 27, 577-584. https://www.cma.gov.cn/en/NewsReleases/MetInstruments/201308/P020130815575473012331.pdf.

Zhang, T., Zhang, R., Zhong, J., Shen, X., Wang, Y., Guo, L. (2023). Classification and estimation of unfavourable boundary-layer meteorological conditions in Beijing for PM2.5 concentration changes using vertical meteorological profiles. *Atmos Res*, 293, 106902. https://doi.org/10.1016/j.atmosres.2023.106902.

The detailed revisions in the manuscript are shown below:

[revised manuscript text omitted]

1. Lines 28-29 of abstract: What is the meaning of OP values? Is 1 nmol/min/m3 low, and ifso, does that mean there is small potential for health risk? Please explain why it is usefulto report the numbers.

**Response**: Thank you very much for your constructive comments. The comments have been answered in detail in the following order:

**First,** following your advice, we have added the definition of OP in the abstract (**Please see Page 2**). The ability of inhaled particles to generate ROS can be quantified by their OP, the ability of specific PM components to deplete antioxidants in assays as a proxy of their in vivo effects (Mylonaki et al., 2024).

**Second,** as mentioned before, Liu et al. (2020) reported $OP^{DTTv}$ values ranging from as high as 14.0 $nmol\ min^{-1}\ m^{-3}$ to as low as 0.15 $nmol\ min^{-1}\ m^{-3}$ across various regions in China. The study (Brehmer et al., 2019) has shown that the range of $OP^{DTTv}$ in Sichuan is between 8.5 $nmol\ min^{-1}\ m^{-3}$ and 10.9 $nmol\ min^{-1}\ m^{-3}$, with an average $OP^{DTTv}$ of 9.6 $nmol\ min^{-1}\ m^{-3}$; $OP^{DTTv}$ values in Beijing ranged from 0.11 $nmol\ min^{-1}\ m^{-3}$ to 0.49 $nmol\ min^{-1}\ m^{-3}$, with an average $OP^{DTTv}$ of 0.19 $nmol\ min^{-1}\ m^{-3}$ (Liu et al., 2014); $OP^{DTTv}$ values in Xi'an ranged from 0.24 $nmol\ min^{-1}\ m^{-3}$ to 1.1 $nmol\ min^{-1}\ m^{-3}$, with an average $OP^{DTTv}$ of 0.51 $nmol\ min^{-1}\ m^{-3}$ (Chen et al., 2019); $OP^{DTTv}$ values in Nanjing ranged from 1.5 $nmol\ min^{-1}\ m^{-3}$ to 3.82 $nmol\ min^{-1}\ m^{-3}$, with an average $OP^{DTTv}$ of 2.42 $nmol\ min^{-1}\ m^{-3}$ (Zhang et al., 2023). Based on the existing literature, there are currently no established criteria for classifying OP values as high or low. Therefore, combining the values of OP data from the existing literature, this study attempts to propose several key values for OP delineation from the distribution interval of OP: <1 $nmol\ min^{-1}\ m^{-3}$, 1 $nmol\ min^{-1}\ m^{-3}$~2 $nmol\ min^{-1}\ m^{-3}$, >2 $nmol\ min^{-1}\ m^{-3}$, which corresponds to low, medium, and high, respectively.

**Third,** generally, lower OP values represent a lower risk of exposure to potentially redox-active aerosols (Liu et al., 2020). Therefore, the OP values reported in this study reflect aerosol toxicity.

Brehmer, C., Lai, A., Clark, S., Shan, M., Ni, K., Ezzati, M., Yang, X., Baumgartner, J., Schauer, J.J., Carter, E. (2019). The Oxidative Potential of Personal and Household PM2.5 in a Rural Setting in Southwestern China. *Environ Sci Technol*, 53, 2788-2798. https://doi.org/10.1021/acs.est.8b05120.

Chen, Q., Wang, M., Wang, Y., Zhang, L., Li, Y., Han, Y. (2019). Oxidative Potential of Water-Soluble Matter Associated with Chromophoric Substances in PM2.5 over Xi'an, China. *Environ Sci Technol*, 53, 8574-8584. https://doi.org/10.1021/acs.est.9b01976.

Liu, Q., Baumgartner, J., Zhang, Y., Liu, Y., Sun, Y., Zhang, M. (2014). Oxidative Potential and Inflammatory Impacts of Source Apportioned Ambient Air Pollution in Beijing. *Environ Sci Technol*, 48, 12920-12929. 10.1021/es5029876.

Liu, Q., Lu, Z., Xiong, Y., Huang, F., Zhou, J., Schauer, J.J. (2020). Oxidative potential of ambient PM2.5 in Wuhan and its comparisons with eight areas of China. *Sci Total Environ*, 701, 134844. https://doi.org/10.1016/j.scitotenv.2019.134844.

Mylonaki, M., Gini, M., Georgopoulou, M., Pilou, M., Chalvatzaki, E., Solomos, S., Diapouli, E., Giannakaki, E., Lazaridis, M., Pandis, S.N., Nenes, A., Eleftheriadis, K., Papayannis, A. (2024). Wildfire and African dust aerosol oxidative potential, exposure and dose in the human respiratory tract. Sci Total Environ, 913, 169683. https://doi.org/10.1016/j.scitotenv.2023.169683.

Zhang, L., Hu, X., Chen, S., Chen, Y., Lian, H. (2023). Characterization and source apportionment of oxidative potential of ambient PM2.5 in Nanjing, a megacity of Eastern China. *Env Pollut Bioavail*, 35, 2175728. https://doi.org/10.1080/26395940.2023.2175728.

The detailed revisions in the manuscript are shown below:

**Abstract** (**Page 2**)**:**

"China has long-term high $PM_{2.5}$ levels, and its Oxidative Potential (OP) is worth studying as it may unravel the impacts of aerosol pollution on public health better than $PM_{2.5}$ alone. OP refers to the ability of $PM_{2.5}$ to induce oxidative stress (OS)."

2. Section 2.2. It was not clear whether DEHM represents any feedbacks of aerosols on themeteorology. Could this be clearly explained in the model description.

**Response**: Thank you very much for your constructive comments. In the DEHM model, the meteorological input files required for both Scenario $C_1$ and Scenario $C_2$ are generated by the WRF driven by the ERA5 reanalysis dataset. The meteorological input files required for Scenario $C_3$ are generated by the WRF driven by global meteorological data from the CESM. The anthropogenic emission inventories required for Scenarios $C_2$ and $C_3$ are sourced from Eclipse v6b, while the anthropogenic emission inventory required for Scenario $C_1$ is sourced from EDGAR-HTAP. The model considers surface layer schemes, boundary layer parameterization schemes, cumulus parameterization schemes, longwave radiation schemes, CAM shortwave radiation schemes, gas-phase chemistry schemes, etc. Following your suggestion, we have provided detailed descriptions of the model configurations in Section 2.3 (**Please see Pages 7-8**).

The detailed revisions in the manuscript are shown below:

**Materials and methods** (**Pages 7-8**)**:**

[revised manuscript text omitted]

3. Equation 1 (line 151) gives the calculation for oxidative potential as a function of $PM_{2.5}$ concentrations. If I understand correctly, this equation comes from Liu et al. (2018) citedin the paper and is based on positive matrix factorization performed on samples from coastal cities. Could an explanation be added stating 1) units of the terms, and 2) the reliability of using this equation outside of an industrial coastal region. To me, this is another uncertainty embedded in this study.

**Response**: Thank you very much for your constructive comments. As you understand, the OP prediction model (Equation (1)) used in this study is based on the work of Liu et al. (2018), which quantifies the contribution of coastal city $PM_{2.5}$ emission sources to $DTT_V$ activity by jointly using Positive Matrix Factorization and Multiple Linear Regression models. This model considers the population density of different coastal cities and the impact of different emission sources. Therefore, we assume that this model provides quantitative contributions of different emission sources to $PM_{2.5}$ mass concentrations, and we apply it to areas outside coastal cities. We have evaluated this model, and the results show that it performs well in areas outside coastal regions too. It should be acknowledged that the model is indeed more suitable for coastal cities. Following your suggestion, we have discussed the reliability of using this equation outside coastal areas in the uncertainty analysis in Section 3.5 **(Please see Pages 24-26)**. We appreciate your kind reminder, and we have added units for terms in Section 2.2 **(Please see Page 6)**.

[revised manuscript text omitted]

4. Line 207. Why is the emissions reduction 30% and not another number? Please justify. Have any tests been done to learn about the response to different assumed emission reduction values? Perhaps this could be included in a section discussing uncertainties of the study's results.

**Response**: Thank you very much for your constructive comments. The choice of 30 % was motivated by the consideration that the perturbation would be large enough to produce a sizeable impact (i.e., more than numerical noise) even at long distances, while small enough to be in the near-linear atmospheric chemistry regime (Im et al., 2019). Based on past testing and experience, it has been demonstrated that setting the emissions reduction at 30% ensures the reliability of the results. Therefore, this study did not discuss the uncertainty of the 30% emission reduction setting. Following your suggestion, we have clarified the reasons for setting the emission reduction value at 30% in Section 2.4.2 **(Please see Page 10)**.

The detailed revisions in the manuscript are shown below:

**Materials and methods** (**Page 10**):

"The emission from each individual source is reduced by 30%. The choice of 30 % was motivated by the consideration that the perturbation would be large enough to produce a sizeable impact (i.e., more than numerical noise) even at long distances, while small enough to be in the near-linear atmospheric chemistry regime (Galmarini et al., 2017; Im et al., 2019)."

Galmarini, S., Koffi, B., Solazzo, E., Keating, T., Hogrefe, C., Schulz, M., Benedictow, A., Griesfeller, J.J., Janssens-Maenhout, G., Carmichael, G., Fu, J., Dentener, F. (2017). Technical note: Coordination and harmonization of the multi-scale, multi-model activities HTAP2, AQMEII3, and MICS-Asia3: simulations, emission inventories, boundary conditions, and model output formats. *Atmos Chem Phys*, 17, 1543-1555. https://doi.org/10.5194/acp-17-1543-2017.

Im, U., Christensen, J.H., Nielsen, O.K., Sand, M., Makkonen, R., Geels, C., Anderson, C., Kukkonen, J., Lopez-Aparicio, S., Brandt, J. (2019). Contributions of Nordic anthropogenic emissions on air pollution and premature mortality over the Nordic region and the Arctic. *Atmos Chem Phys*, 19, 12975-12992. https://doi.org/10.5194/acp-19-12975-2019.

5. Line 207. Could the authors clarify whether only $PM_{2.5}$ emissions are reduced or if any precursors (e.g. SO2) also have emission reductions. If I understand the methods correctly, it appears that only primary $PM_{2.5}$ emissions are perturbed for each sector. Since about half the aerosol is produced by chemistry (line 227) and emission controls can be placed on $PM_{2.5}$ precursors (e.g., SO2 and NOx), it seems that additional calculations for reducing precursor species from different sectors is warranted. Could the authors please justify why only primary $PM_{2.5}$ anthropogenic emission reductions are investigated, or add results presenting the impact of reducing $PM_{2.5}$ precursors on the oxidative potential is examined.

**Response**: Thank you very much for your constructive comments. We implemented uniform emission controls on primary particles and gas emissions from specific emission sectors. Literature (Hodan et al., 2004; Chen et al., 2018; Zhang et al., 2022) has shown that in China, the proportion of secondary and primary $PM_{2.5}$ mass to total $PM_{2.5}$ mass is approximately equal; therefore, we assumed that they each constitute 50% (as presented on Page 11). In this study, by implementing uniform emission controls on primary particles and gas emissions from specific emission sectors, we conducted perturbation simulations for these emission sectors. Subsequently, we estimated the respective contributions of these emission sectors to $PM_{2.5}$ concentration/OP using Equations (9~11). In summary, this study perturbed both primary particles and gas emissions

simultaneously.

Chen, P., Wang, T., Kasoar, M., Xie, M., Li, S., Zhuang, B., Li, M. (2018). Source Apportionment of PM2.5 during Haze and Non-Haze Episodes in Wuxi, China. *Atmosphere (Basel)*, 9, 267. https://doi.org/10.3390/atmos9070267.

Hodan, W.M., Barnard, W.R. (2004). Evaluating the contribution of PM2. 5 precursor gases and re-entrained road emissions to mobile source PM2. 5 particulate matter emissions. *MACTEC Federal Programs, Research Triangle Park, NC*. https://www3.epa.gov/ttnchie1/conference/ei13/mobile/hodan.pdf.

Zhang, H., Li, N., Tang, K., Liao, H., Shi, C., Huang, C., Wang, H., Guo, S., Hu, M., Ge, X. (2022). Estimation of secondary PM 2.5 in China and the United States using a multi-tracer approach. *Atmos Chem Phys*, 22, 5495-5514. https://doi.org/10.5194/acp-2021-683.

6. Lines 218-225. Could information on the fraction of each sub-sector's contribution to E$re$ and E$pp$ be added?

**Response**: Thank you very much for your constructive comments. As introduced in Section 2.4.2, the percentage contribution ($PC_{re\_j}$) of PM$_{2.5}$ emissions from residential subsector j ($E_{re\_j}$) to the total PM$_{2.5}$ emissions from the residential sector ($E_{re}$) is calculated using Equation (6). The values for $E_{re\_j}$ and $E_{re}$ can be obtained from the literature (Yun et al., 2020). After calculation, the values for $PC_{re_{\text{coal cooking}}}$, $PC_{re_{\text{coal heating}}}$, $PC_{re_{\text{biomass cooking}}}$, and $PC_{re_{\text{biomass heating}}}$ are determined to be 21%, 27%, 33%, and 19%, respectively. The percentage contributions ($PC_{pp\_bi\_cf}$) of PM$_{2.5}$ emissions from biomass power plants ($E_{pp\_bi}$) and coal-fired power plants ($E_{pp\_cf}$) to the total PM$_{2.5}$ emissions from power plants ($E_{pp}$) are calculated using Equations (7~8), where $EF$, $FQ$, $E_{pp\_cf}$, and $E_{pp}$ are obtained from the literature (Zheng et al., 2018; Tong et al., 2018; Yun et al., 2020; MEE, 2020; Wang et al., 2020; Tang et al., 2020; Lin et al., 2021; Chen et al., 2022). After calculation, $PC_{pp\_bi\_cf}$ is determined to be 54%. Following your suggestion, we have added the percentage contributions of sub-sectors to $E_{re}$ and $E_{pp}$ in Section 2.4.2 **(Please see Page 11)**.

The detailed revisions in the manuscript are shown below:

**Materials and methods** (**Page 11**):

"The values of $E_{re\_j}$ and $E_{re}$ are obtained from the literature (Yun et al., 2020). After calculation, the values for $PC_{re_{\text{coal cooking}}}$, $PC_{re_{\text{coal heating}}}$, $PC_{re_{\text{biomass cooking}}}$, and $PC_{re_{\text{biomass heating}}}$ are determined to be 21%, 27%, 33%, and 19%, respectively."

"$EF$, $FQ$, $E_{pp\_cf}$, and $E_{pp}$ are obtained from the literature (Zheng et al., 2018; Tong et al., 2018; Yun et al., 2020; MEE, 2020; Wang et al., 2020; Tang et al., 2020; Lin et al., 2021; Chen et al., 2022). After calculation, $PC_{pp\_bi\_cf}$ is determined to be 54%."

Yun, X., Shen, G.F., Shen, H.Z., Meng, W.J., Chen, Y.L., Xu, H.R., Ren, Y., Zhong, Q.R., Du, W., Ma, J.M., Cheng, H.F., Wang, X.L., Liu, J.F., Wang, X.J., Li, B.G., Hu, J.Y., Wan, Y., Tao, S. (2020). Residential solid fuel emissions contribute significantly to air pollution and associated health impacts in China. *Sci Adv*, 6. https://www-science-org.ez.statsbiblioteket.dk/doi/10.1126/sciadv.aba7621.

Zheng, B., Tong, D., Li, M., Liu, F., Hong, C.P., Geng, G.N., Li, H.Y., Li, X., Peng, L.Q., Qi, J., Yan, L., Zhang, Y.X., Zhao, H.Y., Zheng, Y.X., He, K.B., Zhang, Q. (2018). Trends in China's anthropogenic emissions since 2010 as the consequence of clean air actions. *Atmos Chem Phys*, 18, 14095-14111. https://doi.org/10.5194/acp-18-14095-2018.

Tong, D., Zhang, Q., Liu, F., Geng, G., Zheng, Y., Xue, T., Hong, C., Wu, R., Qin, Y., Zhao, H., Yan, L., He, K. (2018). Current Emissions and Future Mitigation Pathways of Coal-Fired Power Plants in China from 2010 to 2030. *Environ Sci Technol*, 52, 12905-12914. https://doi.org/10.1021/acs.est.8b02919.

Yun, X., Shen, G.F., Shen, H.Z., Meng, W.J., Chen, Y.L., Xu, H.R., Ren, Y., Zhong, Q.R., Du, W., Ma, J.M., Cheng, H.F., Wang, X.L., Liu, J.F., Wang, X.J., Li, B.G., Hu, J.Y., Wan, Y., Tao, S. (2020). Residential solid fuel emissions contribute significantly to air pollution and associated health impacts in China. *Sci Adv*, 6. https://www-science-org.ez.statsbiblioteket.dk/doi/10.1126/sciadv.aba7621.

MEE. (2020). Bulletin of the second National Survey of pollution sources. http://www.gov.cn/xinwen/2020-06/10/content_5518391.htm. Accessed 11 January, 2023.

Wang, G., Deng, J., Zhang, Y., Zhang, Q., Duan, L., Hao, J., Jiang, J. (2020). Air pollutant emissions from coal-fired power plants in China over the past two decades. *Sci Total Environ*, 741, 140326. https://doi.org/10.1016/j.scitotenv.2020.140326.

Tang, L., Xue, X., Qu, J., Mi, Z., Bo, X., Chang, X., Wang, S., Li, S., Cui, W., Dong, G. (2020). Air pollution emissions from Chinese power plants based on the continuous emission monitoring systems network. *Sci Data*, 7, 325. https://doi.org/10.1038/s41597-020-00665-1.

Lin, S., Tian, H., Hao, Y., Wu, B., Liu, S., Luo, L., Bai, X., Liu, W., Zhao, S., Hao, J., Guo, Z., Lv, Y. (2021). Atmospheric emission inventory of hazardous air pollutants from biomass direct-fired power plants in China: Historical trends, spatial variation characteristics, and future perspectives. *Sci Total Environ*, 767, 144636. https://doi.org/10.1016/j.scitotenv.2020.144636.

Chen, L., Wang, T., Bo, X., Zhuang, Z., Qu, J., Xue, X., Tian, J., Huang, M., Wang, P., Sang, M. (2022). Thermal Power Industry Emissions and Their Contribution to Air Quality on the Fen-Wei Plain. *Atmosphere (Basel)*, 13. https://doi.org/10.3390/atmos13050652.

7. Equation 9. Could the authors explain the 30% term in this equation? If it is the percentage reduction, then wouldn't this term change depending on which sourcesector is perturbed?

**Response**: Thank you very much for your constructive comments. In this study, perturbation simulations were conducted by implementing a 30% emission reduction ratio for specific emission sources to calculate the corresponding response to a 30% reduction. These responses were then converted into contributions of the emission sources to $PM_{2.5}$ concentration/OP using Equations (9~11). Considering that the implementation of emission reductions may affect atmospheric dynamic processes, we set the emission reduction ratio to 30% based on previous tests and experience. As previously replied, the choice of 30 % was motivated by the consideration that the perturbation would be large enough to produce a sizeable impact (i.e., more than numerical noise) even at long distances, while small enough to be in the near-linear atmospheric chemistry regime (Im et al., 2019). This also indicates that this value will not change due to different perturbed sectors.

Im, U., Christensen, J.H., Nielsen, O.K., Sand, M., Makkonen, R., Geels, C., Anderson, C., Kukkonen, J., Lopez-Aparicio, S., Brandt, J. (2019). Contributions of Nordic anthropogenic emissions on air pollution and premature mortality over the Nordic region and the Arctic. *Atmos Chem Phys*, 19, 12975-12992. https://doi.org/10.5194/acp-19-12975-2019.

8. Figure 3. Are the results shown for simulation C1? Why not also show the evaluation forsimulations C2 and C3? Throughout the manuscript, it would be informative to see the figures for all three scenarios. Please add these figures in either the main manuscript orthe supplement.

**Response**: Thank you for this kind reminding and suggestions. As you understand, Figure 3 presents the model evaluation results for Scenario $C_1$. To comprehensively assess the overall performance of the models under three scenarios, we have supplemented density scatter plots for Scenarios $C_2$ and $C_3$ in the supplemental materials (as shown in Figure S2) as per your suggestion and discussed them in Section 3.1 **(Please see Pages 13-14 in the manuscript and Page 4 in the supplemental materials)**.

The detailed revisions in the manuscript are shown below:

**Results and discussion** (**Pages 13-14**):

"The density scatter plot of model performance and evaluation for China in scenario $C_1$ based on annual mean $PM_{2.5}$ observations from MEE and $PM_{2.5}$ derived from the Dalhousie dataset are shown in Figure 3. Overall, the model performance in terms of correlation coefficient (R) and normalized mean error (NME) calculated based on annual mean observations met the performance criteria suggested by Emery et al. (2017) (NME<0.5, R>0.4), and the normalized mean bias (NMB) was also close to the performance criteria suggested by Emery et al. (2017) (NMB<±0.3). Compared to the observations, the model performance in terms of R, NME, and NMB calculated based on the Dalhousie dataset was slightly poorer but still close to the performance criteria suggested by Emery et al. (2017). Additionally, this study also evaluated the model performance in scenarios $C_2$ and $C_3$, as illustrated in Figures S2. Figures S2a (c) and S2b (d) depicted density scatter plots of model performance and evaluation in scenarios $C_2$ ($C_3$) based on annual mean observations and the Dalhousie dataset, respectively. It was found that under scenarios $C_2$ and $C_3$, the model performance in terms of R and NME, calculated based on both annual mean observations and Dalhousie dataset met the performance criteria suggested by Emery et al. (2017). The NMB under scenarios $C_2$ and $C_3$ calculated based on both annual mean observations and Dalhousie dataset were also close to the performance criteria suggested by Emery et al. (2017). Therefore, the simulated annual mean $PM_{2.5}$ concentrations in scenarios $C_1$, $C_2$ and $C_3$ is considered reliable.

[Figure]

Figure 3. Density scatterplots of model performance and validation for China in scenario $C_1$ based on (a) annual mean $PM_{2.5}$ observations from MEE and (b) annual mean $PM_{2.5}$ derived from the Dalhousie dataset in 2014."

The detailed revisions in the supplemental materials are shown below (**Page 4**):
"

[Figure]

Figure S2. Density scatterplots of model performance and validation in scenario $C_2$ and $C_3$ for China in 2014; (a) and (b) represent the results in scenario $C_2$ based on annual mean $PM_{2.5}$ observations and annual mean $PM_{2.5}$ derived from the Dalhousie dataset, respectively; (c) and (d) represent the results in scenario $C_3$ based on annual mean $PM_{2.5}$ observations and annual mean $PM_{2.5}$ derived from the Dalhousie dataset, respectively."

Emery, C., Liu, Z., Russell, A.G., Odman, M.T., Yarwood, G., Kumar, N. (2017). Recommendations on statistics and benchmarks to assess photochemical model performance. *J Air Waste Manag Assoc*, 67, 582-598. https://doi-org.ez.statsbiblioteket.dk/10.1080/10962247.2016.1265027.

9. Are the Figure 4 results for the annual mean? If so, please state this and explain if it issubtracting the annual mean of the simulation results from the annual mean of the observations. In contrast, it could be differencing individual time points for each location and then taking an annual mean of the difference.

**Response**: Thank you very much for your constructive comments. Figure 4 depicted the spatial distribution of the annual mean simulated minus annual mean observed values, as well as the spatial distribution of the annual mean simulated values minus the Dalhousie dataset. Following your suggestion, we have clarified the content of Figure 4 in Section 3.1 and the figure caption (**Please see Pages 14-15**).

The detailed revisions in the manuscript are shown below:

**Results and discussion:**

"Figure 4 showed the spatial distribution of the annual mean simulated minus annual mean observed values (denoted as $diff_{si-ob}$) (Figure 4a), as well as the spatial distribution of the annual mean simulated values minus the Dalhousie dataset (denoted as $diff_{si-DH}$) (Figure 4b)." (**Page 14**)

"

[Figure]

Figure 4. Spatial distribution of the annual mean simulated minus annual mean observed values (a), as well as the spatial distribution of the annual mean simulated values minus the Dalhousie dataset (b) for China in 2014 under scenario $C_1$." (**Page 15**)

10. Figure 4 shows differences of up to 18 ug/m3 in central China. How does this compare to the annual mean concentrations? It would be helpful to also see a percent differencemap.

**Response**: Thank you very much for your constructive comments. Thank you very much for your suggestion. Figure 4 showed differences as high as -18$\mu g\ m^{-3}$ in central and eastern China, which is an underestimation of 37% compared to the average annual observations. While it is indeed true that supplementing with percentage difference maps could provide a better comparison of the bias between simulated and observed (Dalhousie dataset) values, considering that this study has already divided China into four regions, a comparison from a regional perspective may be more appropriate. Therefore, we have supplemented the bias between simulated and observed (Dalhousie dataset) values for the four regions **(Please see Pages 14-15)**. The simulated PM_{2.5} concentrations in eastern, central, northeastern, and western China were 37%, 21%, -49%, and 41% lower than the observations, respectively; the simulated values were 28%, 3%, 54%, and 48% higher than the Dalhousie dataset, respectively. The disparities in model performance across regions may be attributed to uncertainties in the simulation of meteorological fields, coupled with insufficient consideration of species in the reaction processes within the model. Considering the existing literature (Huang et al., 2021; Jia et al., 2021), it is known that bias within approximately 50% is acceptable. For example, the PM_{2.5} concentrations in East China in 2014 simulated by Jia et al. (2021) was overestimated by 48%. Shi et al. (2021) also reported PM_{2.5} concentrations being overestimated or underestimated by 40% compared to observed values. Hence, the simulated bias in this study falls within an acceptable range, meeting the research requirements.

The detailed revisions in the manuscript are shown below:

**Results and discussion (Pages 14-15):**

"Both Figure 4a and Figure 4b indicated that the majority of regions (central and eastern China) exhibited differences ranging from -18 $\mu g\ m^{-3}$ to 0 $\mu g\ m^{-3}$, which is an underestimation of 37% compared to the average annual observations. The simulated PM_{2.5} concentrations in eastern, central, northeastern, and western China were 37%, 21%, -49%, and 41% lower than the observations, respectively; the simulated values were 28%, 3%, 54%, and 48% higher than the Dalhousie dataset, respectively. The disparities in model performance across regions may be attributed to uncertainties in the simulation of meteorological fields, coupled with insufficient consideration of species in the reaction processes within the model. Considering the

existing literature (Huang et al., 2021; Jia et al., 2021), it is known that bias within approximately 50% is acceptable. For example, the PM$_{2.5}$ concentrations in East China in 2014 simulated by Jia et al. (2021) was overestimated by 48%. Shi et al. (2021) also reported PM$_{2.5}$ concentrations being overestimated or underestimated by 40% compared to observed values. Hence, the simulated bias in this study falls within an acceptable range, meeting the research requirements."

Huang, L., Zhu, Y., Zhai, H., Xue, S., Zhu, T., Shao, Y., Liu, Z., Emery, C., Yarwood, G., Wang, Y., Fu, J., Zhang, K., Li, L. (2021). Recommendations on benchmarks for numerical air quality model applications in China – Part 1: PM2.5 and chemical species. *Atmos Chem Phys*, 21, 2725-2743. https://doi.org/10.5194/acp-21-2725-2021.

Jia, W., Zhang, X. (2021). Impact of modified turbulent diffusion of PM2.5 aerosol in WRF-Chem simulations in eastern China. *Atmos Chem Phys*, 21, 16827-16841. https://doi.org/10.5194/acp-21-16827-2021.

Shi, X., Zheng, Y., Lei, Y., Xue, W., Yan, G., Liu, X., Cai, B., Tong, D., Wang, J. (2021). Air quality benefits of achieving carbon neutrality in China. *Sci Total Environ*, 795, 148784. https://doi.org/10.1016/j.scitotenv.2021.148784.

11. Line 297. I disagree with the statement, "We conclude that the model performs well ….", which is stated just after noting the poor agreement in northwest China (whichis for good reasons). Please qualify this statement. I suggest limiting the remainder of the analysis to all of China except northwest China. See also my comments about performing calculations for specified regions.

**Response**: Thank you very much for your constructive comments. We apologize for drawing unscientific conclusions. Based on your reminder, we have rewritten the conclusion **(Please see Page 15)**. The purpose of plotting Figures S1 and 5 in this study was to assess the performance of the DEHM model in China as a whole on a temporal scale. From Figure 5, it can be observed that the simulated values closely align with the observed values from April to September. However, in other months, there was a slightly poorer alignment between simulated and observed values. Nonetheless, considering the overall performance throughout the year, as analyzed in conjunction with Figure S1, it can be deduced that both the correlation R and NME met the performance criteria suggested by Emery et al. (2017) for all months except December. Furthermore, the results in Figure 4 indicated that the bias across various regions in DEHM is acceptable. Consequently, on an aggregate level for China, the model demonstrates acceptable performance in simulating monthly average PM$_{2.5}$ concentrations.

The detailed revisions in the manuscript are shown below:

**Results and discussion** (**Page 15**)**:**

"From Figure 5, it can be observed that the simulated values closely align with the observed values from April to September. However, in other months, there was a slightly poorer alignment between simulated and observed values. Nonetheless, considering the overall performance throughout the year, as analyzed in conjunction with Figure S1, it can be deduced that both the correlation R and NME met the performance criteria suggested by Emery et al. (2017) for all months except December. Furthermore, the results in Figure 4 indicated that the bias across various regions in DEHM is acceptable. Consequently, on an aggregate level for China, the model demonstrates acceptable performance in simulating monthly average PM$_{2.5}$ concentrations."

Emery, C., Liu, Z., Russell, A.G., Odman, M.T., Yarwood, G., Kumar, N. (2017). Recommendations on statistics and benchmarks to assess photochemical model performance. *J Air Waste Manag Assoc*, 67, 582-598. https://doi-org.ez.statsbiblioteket.dk/10.1080/10962247.2016.1265027.

12. Lines 309-320. Do the maps of PM$_{2.5}$ concentration and OP reflect the anthropogenic emissions map of PM$_{2.5}$? As written, this text does not tell me anything new, but I thinkthe authors were hinting at some useful information in the last few lines. I suggest adding maps of anthropogenic emissions for each sector to support why we see the spatial distribution in Figure 6. Then rewrite these lines to focus on these connections.

**Response**: Thank you very much for your constructive comments. Figure 6 illustrates the spatial distribution of overall PM$_{2.5}$ concentration and overall OP across China. The purpose is to reveal the spatial distribution characteristics of overall PM$_{2.5}$ concentration and overall OP, identifying areas of low and high values for both. Additionally, as suggested by you earlier, we have provided the overall annual average PM$_{2.5}$ concentration and overall annual average OP for different regions in numerical form to more intuitively present the pollution levels and differences across regions. We have also considered your suggestion to provide spatial distribution maps for PM$_{2.5}$ concentration and OP for each sector, and this is discussed in Section 3.4 and presented in Figure 11 **(Please see Pages 22-24)**. Since Section 3.4 focuses on investigating the influence of anthropogenic emission sectors on PM$_{2.5}$ concentration and OP, we discuss the levels of PM$_{2.5}$ and OP for each emission sector here, along with the reasons for spatial variations. Furthermore, following your suggestion, we have emphasized the sources of relevant information in the respective positions of Section 3.2 **(Please see Pages 16-17)**.

The detailed revisions in the manuscript are shown below:

**3.2 Spatial distribution characteristics of PM$_{2.5}$ and OP (Pages 16-17):**

[revised manuscript text omitted]

13. Section 3.4 provides good conclusions and interesting points but does not complete the analysis of their model output to explain the results. Instead of going into detail on each panel shown in Figure 10, I suggest highlighting what is learned. What I learned is that secondary aerosol formation is the primary contributor to PM$_{2.5}$, while biomass burningand industrial sources each contribute moderately to PM$_{2.5}$ and residential and traffic emissions are small (negligible?) contributions. I also learned that residential burning has more of an impact on the oxidative potential because it is weighted more (based on equation1) than the other terms. What I did not learn is why secondary aerosol formation is the largest contributor to PM$_{2.5}$ and oxidative potential. Are precursor emissions much greater than PM$_{2.5}$ emissions? What role do oxidants play in controlling secondary aerosol formation? As a consequence of this result, should studieson source attribution to oxidative potential focus on precursor emissions and their source sectors? I did not see any DEHM results focused on carbonaceous aerosols or metals and the oxidative potential equation does not make use of that information. Although these aerosols are key culprits affecting human health, I do not understand

why the text about biomass combustion, coal burning, and traffic emissions discussthese details without supporting information from the DEHM results.

**Response**: We sincerely apologize for any confusion caused by our insufficiently clear explanations regarding the spatial distribution of contributions from various anthropogenic sources. In response to your suggestion, we have thoroughly revised the discussion of the spatial distribution maps, incorporating information from DEHM model outputs, model configurations, and the OP prediction equations **(Please see Pages 21-24)**. The comments have been answered in detail in the following order:

**First,** we apologize for any confusion you may have had about the contributions of the various emission sectors due to our lack of clarity. To address this, we have swapped the positions of the percentage contribution map (Figure 11) and the spatial distribution maps (Figure 10). By presenting the percentage contribution map first, we clarify the extent of influence of each emission sector on PM$_{2.5}$ concentration and

OP. Subsequently, we explain the reasons for spatial contribution differences through the spatial distribution maps. As depicted in the percentage contribution map, the percentage contributions of secondary aerosol formation, biomass burning, industrial sectors, coal combustion for residential heating, and transportation to $PM_{2.5}$ concentration are 48%, 21%, 21%, 6%, and 4%, respectively. Similarly, the percentage contributions of secondary aerosol formation, biomass burning, coal combustion for residential heating, industrial sectors, and transportation to OP are 58%, 21%, 11%, 9%, and 1%, respectively. Despite the relatively low contributions from coal combustion for residential heating and transportation sectors, attention is still required due to their emission contributions.

**Second,** it can be seen from Figure 11 that the main reason that secondary aerosol formation is the main anthropogenic source of both $PM_{2.5}$ concentrations and OP in China is due to the higher pollution levels, more contributions to mass, and toxicity in the central and eastern regions. Relevant study (Molina et al., 2023) has highlighted the significant contribution of secondary aerosol formation to particle mass and intrinsic OP. In this study, *se* represents the sum of concentrations of secondary organic aerosols (SOA) and secondary inorganic aerosols (SIA), which we define as secondary aerosol formation. The DEHM model does not consider all sources of $PM_{2.5}$ emissions. For example, as outlined in Sections 2.2 and Sections 3.4, the transportation sector only considers tailpipe emissions and coal combustion represents only the part used for residential heating during cold seasons. The contribution results in Figures 10 and 11 are based on the anthropogenic emission sources that we considered. Therefore, further investigation is needed to determine whether precursor emissions are larger than $PM_{2.5}$ emissions, based on a more comprehensive consideration of anthropogenic sources. As understood, the main components of *se* (such as sulfate, nitrate, ammonium, and water-soluble organic carbon) are primarily formed through atmospheric reactions of precursor species (such as $SO_2$, $NO_x$, and volatile organic compounds). This implies that controlling precursor emissions would be beneficial in reducing the concentrations of SOA and SIA. Given the anthropogenic source types considered and the results obtained, enhancing research on precursor emissions and their source sectors in China would be advantageous in reducing OP.

**Third,** following your suggestion, we have comprehensively discussed the spatial contribution differences of various anthropogenic sources by integrating the output results of the DEHM model, model configurations, and relevant information regarding the OP prediction equation **(Please see Pages 21-24)**. Due to Chinese crops, especially corn straw, and power plants are mainly concentrated in central and eastern regions, northeast and part of the western region, as well as the bigger intrinsic OP (Equation (1)), this results in biomass burning becoming the second contribution. In this study, coal combustion refers to coal heating from the residential sector. According to Equation (1), OP is determined by both intrinsic OP and emissions. Although the intrinsic OP weight of coal combustion for local cold-season heating is high, the total emissions from this source are relatively low. Therefore, the contribution of coal combustion for local cold-season heating is smaller than that of secondary aerosols and biomass burning. While our study did not separately analyze the contributions of carbonaceous aerosols or metals, we attempted to explain the main reasons for the contribution of industrial processes based on the simulation results. China's four industrial zones (Liaozhong-South Heavy Industry Base, Beijing-Tianjin-Tangshan Industrial Base, Shanghai-Nanjing-Hangzhou Industrial Base, and Pearl River Delta Light Industry Base) are important contributors to $PM_{2.5}$ and OP emissions from industrial sources. These areas are predominantly steel industry bases, with more $PM_{2.5}$ emissions from industrial sources. We speculate that metals from the steel industry bases are a major factor contributing to the high OP from industrial sources. Furthermore, we surveyed previous studies (Fang et al., 2017; Liu et al., 2018), which confirmed that metals are indeed the primary factors contributing to industrial $PM_{2.5}$/OP emissions. The transportation sector in the DEHM model only considers tailpipe emissions,

[revised manuscript text omitted]

**Organization, Clarity, Technical Comments**

1. Lines 29-31. Please write this sentence more clearly: meteorological conditions contributed 46% and 65% to $PM_{2.5}$ concentrations and oxidative potential, respectively, while anthropogenic emissions contributed 54% and 35% to $PM_{2.5}$ concentrations and oxidative potential, respectively.

**Response**: Thank you very much for your constructive comments. Following your suggestion, we have rewritten the sentence. The detailed revisions in the manuscript are shown below:

"Analysis of the simulations indicate that meteorological conditions contributed 46% and 65% to $PM_{2.5}$ concentrations and OP variability, respectively, while anthropogenic emissions contributed 54% and 35% to $PM_{2.5}$ concentrations and OP variability, respectively." (**Page 2**)

2. Line 87. "combing" should be "combining".

**Response**: Thank you very much for your constructive comments. Following your suggestion, we have modified the term. The detailed revisions in the manuscript are shown below:

"For this, we propose a hybrid approach combining existing observations of OP with a chemistry transport model (CTM)" (**Page 4**)

3. Line 105. Please explain further how the DTT measurements are brought into this study. Do you mean via the positive matrix factorization done in a previous study?

**Response**: Thank you very much for your constructive comments. Liu et al. (2018) first used the $PM_{2.5}$ chemical compositional data set to obtain the quantitative contribution of different emission sources to $PM_{2.5}$ mass concentrations. Then, the contributions of different $PM_{2.5}$ sources acquired from the positive matrix factorization analysis were applied, as the independent variables (in concentration unit), to establish a multiple linear regression equation that accounted for $DTT_v$ activity. Finally, A prediction model of OP was obtained (Equation (1)). In this study, the spatial-scale estimation of OP was estimated by incorporating simulated values of primary and secondary $PM_{2.5}$ concentrations from various anthropogenic sources into Equation (1). Following your suggestion, we have rewritten the sentence. The detailed revisions in the manuscript are shown below:

"From these simulations, the spatial-scale estimation of OP was estimated by incorporating simulated values of primary and secondary $PM_{2.5}$ concentrations from various anthropogenic sources into Equation (1) (see Sect.2.2 for detail)." (**Page 5**)

Liu, W., Xu, Y., Liu, W., Liu, Q., Yu, S., Liu, Y., Wang, X., Tao, S. (2018). Oxidative potential of ambient PM2.5 in the coastal cities of the Bohai Sea, northern China: Seasonal variation and source apportionment. *Environ Pollut*, 236, 514-528. https://doi.org/10.1016/j.envpol.2018.01.116.

4. Section 2. I think it would be good to reorganize the section. Currently section 2.3 interrupts the discussion of model simulations in 2.2 and 2.4. Perhaps the explanation of oxidative potential could be presented first followed by the control and sensitivity simulation configurations.

**Response**: Thank you very much for your constructive comments. Following your suggestion, we first present the estimation of the OP (Sect. 2.2) and then describe the configuration of the DEHM model (Sect. 2.3).

The detailed revisions in the manuscript are shown below (**Please see Pages 6-8**):

[revised manuscript text omitted]

5. Section 2.2, meteorology setup. Please give more detail. I assume the nested domain at 50 km x 50 km horizontal grid spacing was used for the analysis, but this was not stated explicitly. I also assume that the outer domain provided initial and boundary conditions for the nested domain, but it was not stated. Was there a spin-up period for the simulations before conducting the analysis for 2014? How frequently was the meteorology data updated to reanalysis (or nudged to reanalysis)? Why was ERA5 chosen to drive the WRF model and not another global reanalysis product like MERRA orNCEP?

**Response**: Thank you very much for your constructive comments. The comments have been answered in detail in the following order:

**First,** following your suggestion, we have stated in Section 2.3 that the nested domain with a horizontal grid spacing of 50 km × 50 km was employed for analysis. Furthermore, we have elaborated on how the outer domain provided initial conditions and boundary conditions for the nested domain (**Please see Page 7**).

**Second,** to minimize the impact of initial conditions, a spin-up period was configured for the simulations. In this study, the meteorological input files required by the DEHM model were generated using WRF driven by either ERA5 reanalysis datasets or global meteorological data output by CESM. Therefore, the frequency of meteorological data updates to the reanalysis (or nudged to the reanalysis) depends on the specific meteorological data utilized.

**Third,** studies (Thomas et al., 2021; Xu et al., 2022) have demonstrated that ERA5 performs well relative to MERRA, NCEP, and ERA-Interim, with higher temporal and spatial resolutions. Therefore, this study chose ERA5 to drive the WRF model. Considering the sensitivity experiments regarding meteorological conditions and emissions outlined in Section 2.4.1, we elucidate here the reasons for the selection of ERA5 (**Please see Page 8**).

The detailed revisions in the manuscript are shown below:

**Materials and methods:**

"The nested domain covered the whole of China consisting of $150 \times 150$ grid cells with a resolution of 50 km $\times$ 50 km, which was used for the analysis. The mother domain provided initial and boundary conditions for the nested domain." (**Page 7**)

"Table 1 summarizes the scenarios for assessing the relative contributions of meteorological conditions and emissions to $PM_{2.5}$ and OP variability in 2014. ERA5 (Hersbach et al., 2020; ERA, 2023) is a global reanalysis dataset that is based on the assimilation of historical observations and model data. Studies (Thomas et al., 2021; Xu et al., 2022) have demonstrated that ERA5 performs well relative to MERRA, NCEP, and ERA-Interim, with higher temporal and spatial resolutions. Therefore, Scenarios $C_1$ and $C_2$ used ERA5 as input to WRF. Considering the robust representation of aerosol effective radiative forcing and good predictive capabilities for key surface variables in CESM (2023) (García-Martínez et al., 2020; Richter et al., 2022), Scenario $C_3$ utilized meteorological data based on CESM version 2.1.1 (Danabasoglu et al., 2020) climate model as input for WRF. Scenarios $C_2$ and $C_3$ employed the Eclipse V6 emissions inventory, while Scenario $C_1$ used the EDGAR-HTAP inventory." (**Page 8**)

CESM. (2023). Community Earth System Model. https://www.cesm.ucar.edu/. Accessed 7 May, 2023.

Danabasoglu, G., Lamarque, J.F., Bacmeister, J., Bailey, D.A., DuVivier, A.K., Edwards, J., Emmons, L.K., Fasullo, J., Garcia, R., Gettelman, A., Hannay, C., Holland, M.M., Large, W.G., Lauritzen, P.H., Lawrence, D.M., Lenaerts, J.T.M., Lindsay, K., Lipscomb, W.H., Mills, M.J., Neale, R., Oleson, K.W., Otto-Bliesner, B., Phillips, A.S., Sacks, W., Tilmes, S., van Kampenhout, L., Vertenstein, M., Bertini, A., Dennis, J., Deser, C., Fischer, C., Fox-Kemper, B., Kay, J.E., Kinnison, D., Kushner, P.J., Larson, V.E., Long, M.C., Mickelson, S., Moore, J.K., Nienhouse, E., Polvani, L., Rasch, P.J., Strand, W.G. (2020). The Community Earth System Model Version 2 (CESM2). *J Adv Model Earth Syst*, 12, e2019MS001916. https://doi.org/10.1029/2019MS001916.

ERA. (2023). ECMWF Reanalysis v5. https://www.ecmwf.int/en/forecasts/dataset/ecmwf-reanalysis-v5. Accessed 22 May, 2023.

García-Martínez, I.M., Bollasina, M.A., Undorf, S. (2020). Strong large-scale climate response to North American sulphate aerosols in CESM. *Environ Res Lett*, 15, 114051. https://iopscience.iop.org/article/10.1088/1748-9326/abbe45.

Hersbach, H., Bell, B., Berrisford, P., Hirahara, S., Horányi, A., Muñoz-Sabater, J., Nicolas, J., Peubey, C., Radu, R., Schepers, D., Simmons, A., Soci, C., Abdalla, S., Abellan, X., Balsamo, G., Bechtold, P., Biavati, G., Bidlot, J., Bonavita, M., De Chiara, G., Dahlgren, P., Dee, D., Diamantakis, M., Dragani, R., Flemming, J., Forbes, R., Fuentes, M., Geer, A., Haimberger, L., Healy, S., Hogan, R.J., Hólm, E., Janisková, M., Keeley, S., Laloyaux, P., Lopez, P., Lupu, C., Radnoti, G., de Rosnay, P., Rozum, I., Vamborg, F., Villaume, S., Thépaut, J. (2020). The ERA5 global reanalysis. *Q J R Meteorol Soc*, 146, 1999-2049. https://doi.org/10.1002/qj.3803.

Richter, J.H., Glanville, A.A., Edwards, J., Kauffman, B., Davis, N.A., Jaye, A., Kim, H., Pedatella, N.M., Sun, L., Berner, J., Kim, W.M., Yeager, S.G., Danabasoglu, G., Caron, J.M., Oleson, K.W. (2022). Subseasonal Earth System Prediction with CESM2. *Weather Forecast*, 37, 797-815. https://doi.org/10.1175/WAF-D-21-0163.1.

Thomas, S.R., Nicolau, S., Martínez-Alvarado, O., Drew, D.J., Bloomfield, H.C. (2021). How well do atmospheric reanalyses reproduce observed winds in coastal regions of Mexico? *Meteorol Appl*, 28, e2023. https://doi.org/10.1002/met.2023.

Xu, X., Frey, S.K., Ma, D. (2022). Hydrological performance of ERA5 and MERRA-2 precipitation products over the Great Lakes Basin. *Journal of Hydrology: Regional Studies*, 39, 100982. https://doi.org/10.1016/j.ejrh.2021.100982.

6. Section 2.2, chemistry configuration. It would be useful to give a description of the gas-phase chemistry and how $PM_{2.5}$ is formed. For example, what hydrocarbons are included that would contribute to SOA formation? What sulfur chemistry and nitrogen oxides chemistry are represented that make sulfate and nitrate aerosol? Consider including the list of chemical reactions in the supplement. Are the aerosols representedwith a bulk aerosol scheme?

**Response**: Thank you very much for your constructive comments. The gas-phase species considered in this study included $SO_2$, $NO_2$, $CH_4$, $C_2H_6$, etc. $PM_{2.5}$ was formed by BC, OC, sea salt, ammonium ($NH_4^+$), nitrate ($NO_3^-$), sulfate ($SO_4^{2-}$), and secondary organic aerosols (SOA), among others (Frohn et al., 2022). Biogenic volatile organic compounds (BVOCs), such as isoprene, contributed to the formation of SOA (Zare et al., 2012). Considering that the configuration of the specific chemical scheme and the list of chemical reactions have been described in detail in previous work (Zare et al., 2012; Brandt et al., 2012; Collin, 2020; Frohn et al., 2022), in order to respect the efforts and achievements of previous research, appropriate citations have been included in the main text. Additionally, the basic chemical scheme in DEHM was based on the scheme by Strand et al. (1994). In accordance with your suggestion, corresponding modifications have been made to Section 2.3 (**Please see Page 7**).

The detailed revisions in the manuscript are shown below:

**Materials and methods (Page 7):**

"The gas-phase chemistry module included 66 species, 9 primary particles (including natural particles such as sea salt), and 138 chemical reactions and was based on the scheme by Strand et al. (1994) (Brandt et al., 2012). The gas-phase species considered in this study included $SO_2$, $NO_2$, $CH_4$, $C_2H_6$, etc. $PM_{2.5}$ was formed by BC, OC, sea salt, ammonium ($NH_4^+$), nitrate ($NO_3^-$), sulfate ($SO_4^{2-}$), and secondary organic aerosols (SOA), among others (Frohn et al., 2022). Biogenic volatile organic compounds (BVOCs), such as isoprene, contributed to the formation of SOA (Zare et al., 2012). Further details on the configuration of the chemical scheme and the list of chemical reactions can refer to the literature (Zare et al., 2012; Brandt et al., 2012; Collin, 2020; Frohn et al., 2022)."

Strand, A., Hov, Ø. (1994). A two-dimensional global study of tropospheric ozone production. *Journal of Geophysical Research: Atmospheres*, 99, 22877-22895. https://doi.org/10.1029/94JD01945.

Brandt, J., Silver, J.D., Frohn, L.M., Geels, C., Gross, A., Hansen, A.B., Hansen, K.M., Hedegaard, G.B., Skjøth, C.A., Villadsen, H., Zare, A., Christensen, J.H. (2012). An integrated model study for Europe and North America using the Danish Eulerian Hemispheric Model with focus on intercontinental transport of air pollution. *Atmos Environ (1994)*, 53, 156-176. https://doi.org/10.1016/j.atmosenv.2012.01.011.

Collin, G. (2020). Regional Production, Updated documentation covering all Regional operational systems and the ENSEMBLE. https://atmosphere.copernicus.eu/sites/default/files/2020-01/CAMS50_2018SC1_D2.0.2-U1_Models_documentation_201910_v1.pdf. Accessed.

Frohn, L.M., Geels, C., Andersen, C., Andersson, C., Bennet, C., Christensen, J.H., Im, U., Karvosenoja, N., Kindler, P.A., Kukkonen, J., Lopez-Aparicio, S., Nielsen, O., Palamarchuk, Y., Paunu, V., Plejdrup, M.S., Segersson, D., Sofiev, M., Brandt, J. (2022). Evaluation of multidecadal high-resolution atmospheric chemistry-transport modelling for exposure assessments in the continental Nordic countries. *Atmos Environ (1994)*, 290, 119334. https://doi.org/10.1016/j.atmosenv.2022.119334.

Zare, A., Christensen, J.H., Irannejad, P., Brandt, J. (2012). Evaluation of two isoprene emission models for use in a long-range air pollution model. *Atmos Chem Phys*, 12, 7399-7412. https://doi.org/10.5194/acp-12-7399-2012.

7. Section 2.2, emissions. I suggest giving short descriptions of each emissions source: Howare biogenic emissions, sea salt emission, lightning emissions calculated and what are they emitting? Is biomass burning from wildfires included? For anthropogenic emissions,it states EDGAR-HTAP is used, but it does not include a description of what that inventory emits and what grid spacing the inventory has. Yet, in section 2.4 there is a paragraph giving that information for the ECLIPSE emissions. It would be good to have similar information about each inventory so that the reader can better understand why there may be differences between simulation C1 and simulation C2. Especially useful would be to report the emission inventories' annual values for China for PM$_{2.5}$ and key precursors (e.g. SO2) as this will provide quantitative information on how EDGAR-HTAP and Eclipse differ.

**Response**: Thank you very much for your constructive comments. The comments have been answered in detail in the following order:

**First,** following your suggestion, we have provided a brief description of each emission source (**Please see Page 8**).

**Second,** during the simulation process, we included all biomass-burning sources, including wildfires, agricultural burning, and biofuel combustion.

**Third,** following your suggestion, we have added the description of emissions from Eclipse v6 and EDGER-HTAP inventories and grid spacing of the inventories in Section 2.4.1 (**Please see Pages 8-9**).

**Fourth,** your suggestion is excellent. It would be more helpful to provide annual emission inventories for Chinese PM$_{2.5}$ and key precursors (such as sulfur dioxide) under two scenarios. However, these two inventories are from different official sources, and their inventory emission types and values do not correspond one-to-one. This discrepancy results in the PM$_{2.5}$ outcomes being determined by the entire inventory. Reporting only the annual values for Chinese PM$_{2.5}$ and key precursors (such as sulfur dioxide) from these two inventories might give the impression of neglecting the role of other emissions in PM$_{2.5}$ formation. To avoid misunderstandings among readers and considering the difficulty in statistical comparison, we refrained from quantifying specific emissions. Additionally, we have provided inventory links, which include detailed information for each pollutant.

The detailed revisions in the manuscript are shown below:

**Materials and methods:**

[revised manuscript text omitted]

8. Line 152. Please add *se* and its definition.

**Response**: Thank you very much for your constructive comments. Following your suggestion, we have added *se* and its definition. The detailed revisions in the manuscript are shown below:

"where, $re$, $bi$, $in$, and $tr$ represent the primary PM$_{2.5}$ concentrations (with a unit of $\mu g\ m^{-3}$) for coal combustion, biomass burning, industry source, and transportation source, respectively. $se$ (secondary aerosol formation) refers to the concentrations of secondary organic and inorganic (SOA and SIA, respectively) components (with a unit of $\mu g\ m^{-3}$)." (**Page 6**)

9. Line 155. "Industry source is primarily from specific industry processes" is not providing any insight as to what kind of industry or specific types of emissions. Please give more information.

**Response**: Thank you very much for your constructive comments. In this study, the selection and limitation of anthropogenic emission sources are determined by the anthropogenic sources in the OP prediction model (Equation (1)) developed by Liu et al. (2018). Therefore, following your suggestion and Liu et al. (2018).'s research, we have provided additional information regarding industrial sources.

The detailed revisions in the manuscript are shown below:

"Industry source is mainly derived from specific industrial processes in the iron and steel industrial base, metallurgical production plants for non-ferrous metals (e.g., titanium and molybdenum), and so on." (**Page 6**)

Liu, W., Xu, Y., Liu, W., Liu, Q., Yu, S., Liu, Y., Wang, X., Tao, S. (2018). Oxidative potential of ambient PM2.5 in the coastal cities of the Bohai Sea, northern China: Seasonal variation and source apportionment. *Environ Pollut*, 236, 514-528. https://doi.org/10.1016/j.envpol.2018.01.116.

10. Section 2.4.1. There are a number of reanalysis datasets available (e.g., ERA5, MERRA, NCEP FNL). Why were ERA5 and CESM chosen?

**Response**: Thank you very much for your constructive comments. We have elaborated on the reasons for choosing ERA5 and CESM in Section 2.4.1 of the study. The detailed revisions in the manuscript are shown below:

[revised manuscript text omitted]

García-Martínez, I.M., Bollasina, M.A., Undorf, S. (2020). Strong large-scale climate response to North American sulphate aerosols in CESM. *Environ Res Lett*, 15, 114051. https://iopscience.iop.org/article/10.1088/1748-9326/abbe45.

Richter, J.H., Glanville, A.A., Edwards, J., Kauffman, B., Davis, N.A., Jaye, A., Kim, H., Pedatella, N.M., Sun, L., Berner, J., Kim, W.M., Yeager, S.G., Danabasoglu, G., Caron, J.M., Oleson, K.W. (2022). Subseasonal Earth System Prediction with CESM2. *Weather Forecast*, 37, 797-815. https://doi.org/10.1175/WAF-D-21-0163.1.

13. Line 195. I suggest putting the sentence, “abs represents the absolute value” at the end of the paragraph, and rewrite to something more readable, e.g. “In the equations, the abs function represents the absolute value of the quantity in parentheses.”

**Response**: Thank you very much for your constructive comments. Following your suggestion, we have added explanations for abs and the abs function. The detailed revisions in the manuscript are shown below:

“abs represents the absolute value. In the equations, the abs function represents the absolute value of the quantity in parentheses.” (**Page 10**)

14. Line 203. I suggest using “described” instead of “proposed”.

**Response**: Thank you very much for your constructive comments. Following your suggestion, we have used “described” instead of “proposed”. The detailed revisions in the manuscript are shown below:

"These experiments were carried out within the three scenarios described in Section 2.4.1" (**Page 10**)

**15.** Line 225. Change "More and more" to "Previous".

**Response**: Thank you very much for your constructive comments. Following your suggestion, we have changed "More and more" to "Previous". The detailed revisions in the manuscript are shown below:

"Previous studies (Hodan et al., 2004; Chen et al., 2018; Zhang et al., 2022) showed that in China, the proportion of secondary and primary $PM_{2.5}$ mass to the total $PM_{2.5}$ mass is close, so we assume that they account for 50% respectively" (**Page 11**)

Chen, P., Wang, T., Kasoar, M., Xie, M., Li, S., Zhuang, B., Li, M. (2018). Source Apportionment of PM2.5 during Haze and Non-Haze Episodes in Wuxi, China. *Atmosphere (Basel)*, 9, 267. https://doi.org/10.3390/atmos9070267.

Hodan, W.M., Barnard, W.R. (2004). Evaluating the contribution of PM2. 5 precursor gases and re-entrained road emissions to mobile source PM2. 5 particulate matter emissions. *MACTEC Federal Programs, Research Triangle Park, NC*. https://www3.epa.gov/ttnchie1/conference/ei13/mobile/hodan.pdf.

Zhang, H., Li, N., Tang, K., Liao, H., Shi, C., Huang, C., Wang, H., Guo, S., Hu, M., Ge, X. (2022). Estimation of secondary PM 2.5 in China and the United States using a multi-tracer approach. *Atmos Chem Phys*, 22, 5495-5514. https://doi.org/10.5194/acp-2021-683.

**16.** Line 257. Please maintain the same verb tense. I suggest "are mainly … are limited".

**Response**: Thank you very much for your constructive comments. Following your suggestion, we have modified the verb tense. The detailed revisions in the manuscript are shown below:

"In 2014, the observation stations are mainly concentrated in eastern China, while stations in western China are limited." (**Page 13**)

**17.** Line 263. Change "were" to "are".

**Response**: Thank you very much for your constructive comments. Following your suggestion, we have changed "were" to "are". The detailed revisions in the manuscript are shown below:

"The density scatter plot of model performance and evaluation for China in scenario $C_1$ based on annual mean $PM_{2.5}$ observations from MEE and $PM_{2.5}$ derived from the Dalhousie dataset are shown in Figure 3." (**Page 13**)

**18.** Section 3.1. Please specify which model domain is being evaluated.

**Response**: Thank you very much for your constructive comments. Following your advice, we have specified the model domain for evaluation in Section 3.1. The detailed revisions in the manuscript are shown below:

"The density scatter plot of model performance and evaluation for China in scenario $C_1$ based on annual mean $PM_{2.5}$ observations from MEE and $PM_{2.5}$ derived from the Dalhousie dataset are shown in Figure 3." (**Page 13**)

**19.** Line 275. Is it MME or MEE?

**Response**: We sincerely apologize for the oversight in incorrectly stating the data source as "MME." We have now corrected it to "MEE." The detailed revisions in the manuscript are shown below:

"To verify the spatial accuracy, a comparison of the spatial distribution of simulated and observed $PM_{2.5}$, both from MEE and Dalhousie, was conducted." (**Page 14**)

20. Diff_si-ob is used to express the difference between simulated and observed values. Theway this term is written implies that it equals the simulation value minus the observations value. However, the values in Figure 4 appear to be observations minus simulated values. Could the authors clean up the terminology please.

**Response**: Thank you very much for your constructive comments. We have clarified the meaning of $diff_{si-ob}$ in the main text and rewritten the caption for Figure 4 accordingly. The detailed revisions in the manuscript are shown below:

"Figure 4 showed the spatial distribution of the annual mean simulated minus annual mean observed values (denoted as $diff_{si-ob}$) (Figure 4a), as well as the spatial distribution of the annual mean simulated values minus the Dalhousie dataset (denoted as $diff_{si-DH}$) (Figure 4b)." (**Page 14**)

[Figure]

Figure 4. Spatial distribution of the annual mean simulated minus annual mean observed values (a), as well as the spatial distribution of the annual mean simulated values minus the Dalhousie dataset (b) for China in 2014 under scenario $C_1$." (**Page 15**)

21. Line 293. Change "are" to "were.

**Response**: Thank you very much for your constructive comments. In accordance with your previous comments, we have rewritten the section discussing the performance of the model on a temporal scale. The detailed revisions in the manuscript are shown below:

"Similarly, the model performance over time scales was also investigated. Scatter density plots and distribution characteristics of monthly average observations and simulations for all monitoring sites in 2014 were depicted in Figure S1 and Figure 5, respectively. From Figure 5, it can be observed that the simulated values closely align with the observed values from April to September. However, in other months, there was a slightly poorer alignment between simulated and observed values. Nonetheless, considering the overall performance throughout the year, as analyzed in conjunction with Figure S1, it can be deduced that both the correlation R and NME met the performance criteria suggested by Emery et al. (2017) for all months except December. Furthermore, the results in Figure 4 indicated that the bias across various regions in DEHM is acceptable. Consequently, on an aggregate level for China, the model demonstrates acceptable performance in simulating monthly average PM$_{2.5}$ concentrations." (**Page 15**)

Emery, C., Liu, Z., Russell, A.G., Odman, M.T., Yarwood, G., Kumar, N. (2017). Recommendations on statistics and benchmarks to assess photochemical model performance. *J Air Waste Manag Assoc*, 67, 582-598. https://doi-org.ez.statsbiblioteket.dk/10.1080/10962247.2016.1265027.

**22.** Line 293-294. Remove "with Figure S1a … December".

**Response**: Thank you very much for your constructive comments. Following your advice, we have removed "with Figure S1a … December". The detailed revisions in the manuscript are shown below:

"Scatter density plots and distribution characteristics of monthly average observations and simulations for all monitoring sites in 2014 were depicted in Figure S1 and Figure 5, respectively." (**Page 15**)

**23.** Line 306-307. First sentence needs to be written better to something like: To learn about the spatial distributions of $PM_{2.5}$ concentrations and OP, we plot maps of surface$PM_{2.5}$ and OP for scenario C1 (Figure 6).

**Response**: Thank you very much for your constructive comments. Following your advice, we have rewritten the first sentence. The detailed revisions in the manuscript are shown below:

"To learn about the spatial distributions of $PM_{2.5}$ concentrations and OP, we plot maps of surface $PM_{2.5}$ and OP for scenario $C_1$ (Figure 6a and 6b) and quantified the average annual $PM_{2.5}$ concentrations and OP across different regions of China (Figure 6c)." (**Page 16**)

**24.** Line 308. The sentence, "The findings … and OP" is not needed.

**Response**: Thank you very much for your constructive comments. Following your advice, we have removed this sentence.

**25.** Line 313. Is the term "urban areas" for low OP meant? Or is this area more rural?

**Response**: Thank you very much for your constructive comments. Based on your previous comments, we have rewritten this conclusion. The detailed revisions in the manuscript are shown below:

"Low $PM_{2.5}$ concentrations and Low OP are mainly distributed in northeastern and western China." (**Page 16**)

**26.** Line 318. "northern residents in China right region" does not make sense to me.

**Response**: Thank you very much for your constructive comments. Following your advice, we have removed this sentence. The detailed revisions in the manuscript are shown below:

"Due to high population density, socio-economic activities and winter heating needs, large amounts of anthropogenic emissions, especially from industry, transportation, coal burning and biomass burning, exacerbate $PM_{2.5}$ and redox active component pollution." (**Pages 16-17**)

**27.** Lines 336-340. I do not think so many significant digits are needed. I suggest using 85% instead of 84.8%, and likewise for the other numbers used here.

**Response**: Thank you very much for your constructive comments. Following your suggestion, we have reduced the number of significant digits. The detailed revisions in the manuscript are shown below:

"The PDF and CDF results showed that 85% of the total area was above the primary concentrations limit and 40% was above the secondary concentrations limit. In addition, 36% of regions in China have an OP below $1.00\ nmol\ min^{-1}\ m^{-3}$, 41% have an OP between 1.00 and $2.00\ nmol\ min^{-1}\ m^{-3}$, and 23% have an OP above $2.00\ nmol\ min^{-1}\ m^{-3}$." (**Page 18**)

**28.** Line 349. Change to "illustrates". Line 351. Change to "presents".

**Response**: Thank you very much for your constructive comments. Based on your suggestion, we have adjusted the tense of the corresponding verbs accordingly. The detailed revisions in the manuscript are shown below:

"Figure 8 illustrates the spatial distribution maps of $PM_{2.5}$ concentrations and OP under scenarios $C_1$, $C_2$, and $C_3$. Figure 9a presents the annual average $PM_{2.5}$ concentrations and OP under different scenarios, and Figure 9b shows the relative contributions of meteorological conditions and emission inventories." (**Page 19**)

29. Line 350-351. The sentence is not needed as it repeats the figure caption.

**Response**: Thank you very much for your constructive comments. Following your advice, we have removed this sentence.

30. Line 392. Change "are" to "is".

**Response**: Thank you very much for your constructive comments. Considering your previous comments, we have rewritten this conclusion. The detailed revisions in the manuscript are shown below:

"It can be seen from Figure 11 that the main reason that secondary aerosol formation is the main anthropogenic source of both $PM_{2.5}$ concentrations and OP in China is due to the higher pollution levels, more contributions to mass, and toxicity in the central and eastern regions." (**Page 22**)

31. Line 430 and line 432. I recommend reducing the number of significant digits.

**Response**: Thank you very much for your constructive comments. Following your suggestion, we have reduced the number of significant digits. The detailed revisions in the manuscript are shown below:

"Secondary aerosol formation, biomass burning, industrial, coal combustion for residential heating, and transportation sources contributed 48%, 21%, 21%, 6% and 4% to $PM_{2.5}$, respectively. Secondary aerosol formation, biomass burning, coal combustion for residential heating, industrial sources, and transportation sources contributed 58%, 21%, 11%, 9% and 1% to OP, respectively. This means that secondary aerosol formation and biomass burning are the main sources of $PM_{2.5}$ and OP." (**Page 21**)

1. In all figure captions that show results, please include information on the time periodshown (e.g., annual average) and spatial region shown (where appropriate).

**Response**: Thank you very much for your constructive comments. Following your suggestion, we have added the study time, region, and scenarios to the titles of the figures and tables. The detailed revisions in the manuscript are shown below:

[revised manuscript text omitted]

"Figure S1. Density scatterplots of model performance and validation based on monthly mean $PM_{2.5}$ observations for China in 2014 under scenario $C_1$; (a) to (l) are the results from January to December." (**Page 3** in the supplemental materials)

"Figure S2. Density scatterplots of model performance and validation in scenario $C_2$ and $C_3$ for China in 2014; (a) and (b) represent the results in scenario $C_2$ based on annual mean $PM_{2.5}$ observations and annual mean $PM_{2.5}$ derived from the Dalhousie dataset, respectively; (c) and (d) represent the results in scenario $C_3$ based on annual mean $PM_{2.5}$ observations and annual mean $PM_{2.5}$ derived from the Dalhousie dataset, respectively." (**Page 4** in the supplemental materials)

2. Figure 3, figure caption. Please state what parameter is being plotted. I assume $PM_{2.5}$, but it should be explicitly stated.

**Response**: Thank you very much for your constructive comments. Following your suggestion, we have clarified the parameters being plotted in the title of Figure 3. The detailed revisions in the manuscript are shown below:

"Figure 3. Density scatterplots of model performance and validation for China in scenario $C_1$ based on (a) annual mean $PM_{2.5}$ observations from MEE and (b) annual mean $PM_{2.5}$ derived from the Dalhousie dataset in 2014." (**Page 14**)

3. Figure 3 would benefit from having less white space. I suggest changing the maximum value to 120 or 150 ug/m3.

**Response**: Thank you very much for your constructive comments. Following your suggestion, we have changed the maximum value to 150 $\mu g\ m^{-3}$. The detailed revisions in the manuscript are shown below:
"

[Figure]

Figure 3. Density scatterplots of model performance and validation for China in scenario $C_1$ based on (a) annual mean $PM_{2.5}$ observations from MEE and (b) annual mean $PM_{2.5}$ derived from the Dalhousie dataset in 2014." (**Page 14**)

4. Figure 3. Are the points shown in panel b for the Dalhousie dataset for the same locations as the MEE observations? Or are there more points taking advantage of thegridded dataset?

**Response**: Thank you very much for your constructive comments. The points shown in Figure b are different from the MEE observation stations. The data points displayed in Figure b are extracted from the Dalhousie dataset within the Chinese region. In 2014, the observation stations are mainly concentrated in eastern China, while stations in western China are limited. Therefore, in the present study, we also evaluated with the gridded annual-mean global reanalysis Dalhousie surface $PM_{2.5}$ dataset, which combines satellite retrievals of aerosol optical depth, chemical transport modeling, and ground-based measurements. The Dalhousie dataset compensated for the non-uniform distribution spatially of observation stations to comprehensively evaluate the performance of the DEHM model.

5. Figure 4. Adjust the colorbar so that the whitest color is zero. That makes it easier to seedifferences between positive and negative values.

**Response**: Thank you very much for your constructive comments. Following your suggestion, we attempted to adjust the color bar so that the whitest color corresponds to zero. However, the effect was not satisfactory. This is mainly because of the significant difference between the maximum value (positive) and the minimum value (negative). The absolute values of negative values are large, while positive values are small.  In this situation, it is challenging to discern the differences between regions. To ensure that readers can more intuitively perceive the deviations across different regions of China, we have chosen to retain the original version. Once again, we appreciate your advice.

6. Figure 5. Are the observations shown in the figure from the MEE data or the Dalhousiereanalysis? Please note this in the figure caption.

**Response**: Thank you very much for your constructive comments. Following your suggestion, we have added a note in the figure caption indicating that the observed values are from the MEE data. The detailed revisions in the manuscript are shown below:

"Figure 5. Violin plots of monthly average from MEE observations and simulations averaged over various observation stations for China in 2014 under scenario $C_1$; The red and blue colors represent the statistical

distribution of simulated and observations, respectively; The width of the violin represents the sample size; The solid black line inside the violin indicates the median. The upper and lower dashed black lines within the violin indicate the upper quartile (the 75th percentile) and lower quartile (the 25th percentile), respectively." (**Page 16**)

7. Figure 5. It is difficult to discern the dashed and solid horizontal lines because the solidline does not extend from one edge of the colored region to the other. Is it possible tofix this?

**Response**: Thank you very much for your constructive comments. Following your suggestion, we have extended the solid lines from one edge of the colored region to the other edge. The detailed revisions in the manuscript are shown below:
"

[Figure]

Figure 5. Violin plots of monthly average from MEE observations and simulations averaged over various observation stations for China in 2014 under scenario $C_1$; The red and blue colors represent the statistical distribution of simulated and observations, respectively; The width of the violin represents the sample size; The solid black line inside the violin indicates the median. The upper and lower dashed black lines within the violin indicate the upper quartile (the 75th percentile) and lower quartile (the 25th percentile), respectively.**"** (**Page 16**)

8. Figure 8. Using 2 rows and 3 columns makes for smaller panels. I suggest using 3 rowsand 2 columns (transposing the panels). I also suggest adding titles for each column, "PM$_{2.5}$ (units)" and "OP (units)" and then panel labels that simply are the simulation name.

**Response**: Thank you very much for your constructive comments. Following your suggestion, we arranged the spatial distribution maps of annual average PM$_{2.5}$ concentration and annual average OP for different scenarios in China for the year 2014 in a layout of 3 rows and 2 columns. We have also added titles to each column. The detailed revisions in the manuscript are shown below:
"

[Figure]

Figure 8. Spatial distribution of annual mean PM$_{2.5}$ concentrations and annual mean OP for China in 2014 under different scenarios; (a)~(c) are PM$_{2.5}$ concentrations in scenarios $C_1$, $C_2$ and $C_3$, respectively; (d)~(f) is the OP in scenarios $C_1$, $C_2$, and $C_3$, respectively; The meteorological datasets (emission inventories) employed for scenarios $C_1$, $C_2$, and $C_3$ are ERA5 (EDGAR-HTAP), ERA5 (Eclipse V6), and CESM (Eclipse V6), respectively." (**Page 20**)

9. Figure 8. What are the insets showing in the bottom right of each panel? They are notdiscussed, so I suggest removing them.

**Response**: Thank you very much for your constructive comments. This study was conducted on the entire China region, and the inset maps in the bottom right corner of each panel represent zoomed-in portions of the China map. This is a standardized approach for studies involving the map of China, and its presence is necessary. Therefore, we have retained the inset maps in the bottom right corner. We appreciate your comment.

10. Figure 10. Like Figure 8, I suggest using 5 rows and 2 columns instead of 4 rows and 3 columns. I also suggest adding titles for each column, "PM$_{2.5}$ (units)" and "OP (units)"and then panel labels are the sector source (e.g., residential heating).

**Response**: Thank you very much for your constructive comments. Following your suggestion, we have organized the spatial distribution maps of annual average PM$_{2.5}$ concentrations and annual average OP for

different sectors in China for the year 2014 into a layout of 5 rows and 2 columns. Additionally, we have added titles for each column. The detailed revisions in the manuscript are shown below:

"

[Figure]

Figure 11. Spatial distribution of annual mean PM$_{2.5}$ concentrations and annual mean OP from different anthropogenic sources for China in 2014 under scenario C$_1$; (a)~(e) are PM$_{2.5}$ concentrations derived from

coal combustion for residential heating, biomass burning, secondary aerosol formation, industry, and traffic respectively; (f)~(j) are the OP derived from coal combustion for residential heating, biomass burning, secondary aerosol formation, industry, and traffic respectively." (**Pages 23-24**)

11. Figure 11. A more complete figure caption is needed: Percent contribution of different anthropogenic sources (traffic, industry, secondary aerosol formation, biomass burning,coal combustion) to total PM$_{2.5}$ concentration and oxidation potential.

**Response**: Thank you very much for your constructive comments. Following your suggestion, we have refined the captions for the figures. The detailed revisions in the manuscript are shown below:

"Figure 10. Percentage contribution of different anthropogenic sources (coal combustion for residential heating, biomass burning, secondary aerosol formation, industry, and traffic) to total PM$_{2.5}$ concentrations and OP for China in 2014 under scenario C$_1$." (**Page 22**)

**References**

Yang, F., C. Liu, H. Qian, Comparison of indoor and outdoor oxidative potential of PM2.5: pollution levels, temporal patterns, and key constituents, *Environment International*, **155**, 2021, https://doi.org/10.1016/j.envint.2021.106684.

---

## Author Response (AR2)

Manuscript: egusphere-2023-2615

Title: Impact of Meteorology and Aerosol Sources on $PM_{2.5}$ and Oxidative Potential Variability and Levels in China

We appreciate all the valuable comments from the reviewers and editor, which significantly improved the quality of the manuscript. We have studied the comments carefully and revised the manuscript accordingly. The comments and our responses point-to-point are listed below. Changes to the paper are shown in blue, so that the reviewers can easily review them again.

Replies to Reviewers and Editor

First of all, we thank both reviewers and editor for their positive and constructive comments and suggestions.

Reviewers' comments:

**Anonymous referee #2:**
I appreciate that the authors have addressed my comments well. There are a few minor word changes that I would like to suggest before the paper is accepted by the journal.

1. Thank you for providing information of high versus low values of OP. I recognize that there is not an agreed upon number above which is high or low, but it would help the reader if the paper gave some context of previously known values. Therefore, please revise the sentence in lines 28-29 of abstract to the following.
Furthermore, the probability density function revealed that about 40% of areas in China had an annual average $PM_{2.5}$ concentrations exceeding the Chinese concentrations limit. For OP, 36% of the regions have OP below 1 $nmol\ min^{-1}\ m^{-3}$, 41% have OP between 1 and 2 $nmol\ min^{-1}\ m^{-3}$, and 23% have OP above 2 $nmol\ min^{-1}\ m^{-3}$, which are in line with previous measurement studies.
**Response**: Thank you very much for your constructive comments. Following your suggestion, we incorporated the recommended content into lines 28-29 of abstract.
The detailed revisions in the manuscript are shown below:
**Abstract (Lines 28-29):**
"Furthermore, the probability density function revealed that about 40% of areas in China had an annual average $PM_{2.5}$ concentrations exceeding the Chinese concentrations limit. For OP, 36% of the regions have OP below 1 $nmol\ min^{-1}\ m^{-3}$, 41% have OP between 1 and 2 $nmol\ min^{-1}\ m^{-3}$, and 23% have OP above 2 $nmol\ min^{-1}\ m^{-3}$, which are in line with previous measurement studies."

2. Line 47. I found the response to my first major comment, where the authors provide OP results from different locations and season to be very useful. However, I did not see those sentences in the revised manuscript. It would be good to add that information before the last sentence of the first paragraph of the Introduction.
**Response**: Thank you very much for your constructive comments. Following your suggestion, we have added the OP results from different locations and season in the first paragraph of the Introduction.
The detailed revisions in the manuscript are shown below:
**Introduction (Lines 45-49):**
"Liu et al. (2020) summarized OP measurements conducted in nine regions of China around 2014. The results

showed that the average OP content in northern Beijing was highest during the winter of 2016 (~14.0 $nmol\ min^{-1}\ m^{-3}$), while the average OP level in Shanghai during the spring of 2016 was lowest (~0.15 $nmol\ min^{-1}\ m^{-3}$). However, there is currently no exact threshold division of OP values."

Liu, Q., Lu, Z., Xiong, Y., Huang, F., Zhou, J., Schauer, J.J. (2020). Oxidative potential of ambient $PM_{2.5}$ in Wuhan and its comparisons with eight areas of China. *Sci Total Environ*, 701, 134844. https://doi.org/10.1016/j.scitotenv.2019.134844.

3. Line 168. I now understand that DEHM does not include aerosol impacts on radiation or clouds. Please state this in the sentence here. For example:
The current version of the DEHM model does not include wind-blown, resuspended dust emissions or road dust nor aerosol-radiation or radiation-cloud interactions.
As a side note, the high $PM_{2.5}$ concentrations in the PBL affects the amount of solar radiation reaching the surface. Consequently, surface temperatures and PBL height are modified. Further, the aerosols affect the photolysis rates of the trace gas chemistry, impacting oxidant concentrations and therefore the yield of secondary aerosol formation. This information does not need to be included in the manuscript, but it illuminates areas of model improvement that can further refine estimates of OP.
**Response**: Thank you very much for your constructive comments. Following your suggestion, we have added the content that the current version of the DEHM does not include aerosol impacts on radiation or clouds.
The detailed revisions in the manuscript are shown below:
**Materials and methods (Lines 172-174):**
"The current version of the DEHM model does not include wind-blown, resuspended dust emissions or road dust nor aerosol-radiation or radiation-cloud interactions."

4. Line 215. Please remove "abs represents the absolute value."
**Response**: Thank you very much for your constructive comments. Following your suggestion, we have removed "abs represents the absolute value."
The detailed revisions in the manuscript are shown below:
**Materials and methods (Lines 222-224):**
"Con(Emi) represents the impact of changing emission inventory on changes in $PM_{2.5}$ and OP. NCon(Met) and NCon(Emi) represent the normalized contributions of meteorology and emission. In the equations, the abs function represents the absolute value of the quantity in parentheses."

5. Line 219. Please remove "abs represents the absolute value."
**Response**: Thank you very much for your constructive comments. Following your suggestion, we have removed "abs represents the absolute value."
The detailed revisions in the manuscript are shown below:
**Materials and methods (Lines 222-224):**
"Con(Emi) represents the impact of changing emission inventory on changes in $PM_{2.5}$ and OP. NCon(Met) and NCon(Emi) represent the normalized contributions of meteorology and emission. In the equations, the abs function represents the absolute value of the quantity in parentheses."

6. Line 231. Please revise the sentence to say the following.
The emissions of both primary aerosols and tracegases from each individual source are reduced by 30%.
**Response**: Thank you very much for your constructive comments. Following your suggestion, we have modified "The emission from each individual source is reduced by 30%." to " The emissions of both primary aerosols and

tracegases from each individual source are reduced by 30%." (**Please see Lines 235-236**)

7. Line 311. Please revise the sentence to read more easily.

Additionally, this study evaluated the model performance in scenarios C2 and C3 (Figure S2).

**Response**: Thank you very much for your constructive comments. Following your suggestion, we have modified " Additionally, this study also evaluated the model performance in scenarios $C_2$ and $C_3$, as illustrated in Figures S2." to " Additionally, this study evaluated the model performance in scenarios $C_2$ and $C_3$ (Figure S2). " (**Please see Lines 314-315**)

8. Line 312-313. Please remove the following sentence.

Figures S2a (c) and S2b (d) depicted density scatter plots of model performance and evaluation in scenarios C2 (C3) based on annual mean observations and the Dalhousie dataset, respectively.

**Response**: Thank you very much for your constructive comments. Following your suggestion, we have removed "Figures S2a (c) and S2b (d) depicted density scatter plots of model performance and evaluation in scenarios $C_2$ (C3) based on annual mean observations and the Dalhousie dataset, respectively."

The detailed revisions in the manuscript are shown below:

**Results and discussion (Lines 314-317):**

"Additionally, this study evaluated the model performance in scenarios $C_2$ and $C_3$ (Figure S2). It was found that under scenarios $C_2$ and $C_3$, the model performance in terms of R and NME, calculated based on both annual mean observations and Dalhousie dataset met the performance criteria suggested by Emery et al. (2017)."

9. Line 318. The sentence needs a plural verb.

Therefore, the simulated annual mean $PM_{2.5}$ concentrations in scenarios C1, C2 and C3 are considered reliable.

**Response**: Thank you very much for your constructive comments. Following your suggestion, we have modified " Therefore, the simulated annual mean $PM_{2.5}$ concentrations in scenarios $C_1$, $C_2$ and $C_3$ is considered reliable." to " Therefore, the simulated annual mean $PM_{2.5}$ concentrations in scenarios $C_1$, $C_2$ and $C_3$ are considered reliable." (**Please see Lines 319-320**)

10. Figure 4. I realize that my confusion before on the labeling of the Figure 4 panels, i.e. the difference of simulation minus observations, was caused by red colors being negative and blue being positive. Normally it is the other way.

**Response**: Thank you very much for your constructive comments. Thank you for your reminder, and to avoid any misunderstandings, we have swapped the color bars in Figure 4 so that red colors show high values and blue colors show low values.

The detailed revisions in the manuscript are shown below:

**Results and discussion (Lines 343-345):**

"

[Figure]

Figure 4. Spatial distribution of the annual mean simulated minus annual mean observed values (a), as well as the spatial distribution of the annual mean simulated values minus the Dalhousie dataset (b) for China in 2014 under scenario $C_1$.
"